# Adversarial Regression
# with Doubly Non-negative Weighting Matrices

**Tam Le**[*]
RIKEN AIP
tam.le@riken.jp

**Truyen Nguyen**[*]
University of Akron
tnguyen@uakron.edu

**Makoto Yamada**
Kyoto University and RIKEN AIP
makoto.yamada@riken.jp

**Jose Blanchet**
Stanford University
jose.blanchet@stanford.edu

**Viet Anh Nguyen**
Stanford University and VinAI Research
v.anhnv81@vinai.io

## Abstract

Many machine learning tasks that involve predicting an output response can be solved by training a weighted regression model. Unfortunately, the predictive power of this type of models may severely deteriorate under low sample sizes or under covariate perturbations. Reweighting the training samples has aroused as an effective mitigation strategy to these problems. In this paper, we propose a novel and coherent scheme for kernel-reweighted regression by reparametrizing the sample weights using a doubly non-negative matrix. When the weighting matrix is confined in an uncertainty set using either the log-determinant divergence or the Bures-Wasserstein distance, we show that the adversarially reweighted estimate can be solved efficiently using first-order methods. Numerical experiments show that our reweighting strategy delivers promising results on numerous datasets.

## 1   Introduction

We are interested in learning a parameter $\beta$ that has a competitive predictive performance on a response variable $Y$. Given $N$ training samples $(\widehat{z}_i, \widehat{x}_i, \widehat{y}_i)_{i=1}^N$ in which $(\widehat{z}_i, \widehat{x}_i)$ are the contexts that possess explanatory power on $\widehat{y}_i$, learning the parameter $\beta$ can be posed as a weighted regression problem of the form

$$\min_{\beta} \ \sum_{i=1}^N \omega(\widehat{z}_i)\ell(\beta, \widehat{x}_i, \widehat{y}_i). \tag{1}$$

In problem (1), $\omega$ is a weighting function that indicates the contribution of the sample-specific loss to the objective. By aligning the covariate $(\widehat{z}_i, \widehat{x}_i)$ appropriately to the weighting term $\omega(\widehat{z}_i)$ and the loss term $\ell(\beta, \widehat{x}_i, \widehat{y}_i)$, the generic formulation of problem (1) can be adapted to many popular learning and estimation tasks in machine learning. For example, problem (1) encapsulates the family of kernel smoothers, including the Nadaraya-Watson estimator [16, 22, 39].

**Example 1.1** (Nadaraya-Watson (NW) estimator for conditional expectation). *Given the samples* $(\widehat{z}_i, \widehat{y}_i)_{i=1}^N$, *we are interested in estimating the conditional expectation of* $Y$ *given* $Z = z_0$ *for some covariate* $z_0 \in \mathcal{Z}$. *The NW estimator is the optimizer of problem* (1) *with* $\ell(\beta, y) = \|\beta - y\|_2^2$ *and the weighting function* $\omega$ *is given through a kernel* $K$ *via* $\omega(\widehat{z}_i) = K(z_0, \widehat{z}_i)$. *The NW estimate of* $\mathbb{E}[Y|Z = z_0]$ *admits a closed form expression*

$$\beta_{\mathrm{NW}} = \frac{\sum_{i=1}^N K(z_0, \widehat{z}_i)\widehat{y}_i}{\sum_{i=1}^N K(z_0, \widehat{z}_i)}.$$

---

[*]: equal contribution

35th Conference on Neural Information Processing Systems (NeurIPS 2021).

The NW estimator utilizes a locally constant function to estimate the conditional expectation $\mathbb{E}[Y|Z = z_0]$. Locally linear regression [3, 32] extends the NW estimator to reduce the noise produced by the linear component of a target function [27, §3.2].

**Example 1.2** (Locally linear regression (LLR)). *For univariate output and $z \equiv x$, the LLR minimizes the kernel-weighted loss with $\ell([\beta_1, \beta_2], z, y) = (\beta_1 + \beta_2^\top z - y)^2$. The LLR estimate of $\mathbb{E}[Y|Z = z_0]$ admits a closed form expression*

$$\beta_{\text{LLR}} = \left( \left( \widehat{Z}^\top W \widehat{Z} \right)^{-1} \widehat{Z}^\top W \widehat{Y} \right)^\top \begin{bmatrix} 1 \\ z_0 \end{bmatrix},$$

*with $\widehat{Y} = [\widehat{y}_1, \ldots, \widehat{y}_n]^\top \in \mathbb{R}^n$, $W = \text{diag}\left( K(z_0, \widehat{z}_1), \ldots, K(z_0, \widehat{z}_n) \right) \in \mathbb{R}^{n \times n}$ and*

$$\widehat{Z} = \begin{bmatrix} 1 & (\widehat{z}_1 - z_0)^\top \\ \vdots & \vdots \\ 1 & (\widehat{z}_n - z_0)^\top \end{bmatrix}.$$

Intuitively, the NW and LLR estimators are special instances of the larger family of local polynomial estimators with order zero and one, respectively. Problem (1) is also the building block for local learning algorithms [6], density ratio estimation [5, pp.152], risk minimization with covariate shift [17, §4], domain adaptation [36], geographically weighted regression [7], local interpretable explanations [31], to name a few.

In all of the aforementioned applications, a prevailing trait is that the weight $\omega$ is given through a kernel. To avoid any confusion in the terminologies, it is instructive to revisit and distinguish the relevant definitions of kernels. The first family is the non-negative kernels, which are popularly employed in nonparametric statistics [37].

**Definition 1.3** (Non-negative kernel). *A function $K : \mathcal{Z} \times \mathcal{Z} \rightarrow \mathbb{R}$ is non-negative if $K(z, z') \geq 0$ for any $z, z' \in \mathcal{Z}$.*

In addition, there also exists a family of positive definite kernels, which forms the backbone of kernel machine learning [4, 33].

**Definition 1.4** (Positive definite kernel). *A symmetric function $K : \mathcal{Z} \times \mathcal{Z} \rightarrow \mathbb{R}$ is positive definite if for any $n \in \mathbb{N}$ and any choices of $(z_i)_{i=1}^n \in \mathcal{Z}$ and $(\alpha_i)_{i=1}^n \in \mathbb{R}$, we have*

$$\sum_{i=1}^n \sum_{j=1}^n \alpha_i \alpha_j K(z_i, z_j) \geq 0. \tag{2}$$

*Moreover, $K$ is strictly positive definite if we have in addition that for mutually distinct $(z_i)_{i=1}^n \in \mathcal{Z}$, the equality in (2) implies $\alpha_1 = \ldots = \alpha_n = 0$.*

Positive definite kernels are a powerful tool to model geographical interactions [7], to characterize the covariance structure in Gaussian processes [29, §4], and to construct non-linear kernel methods [33]. Interestingly, the two above-mentioned families of kernels have a significant overlap. Examples of kernels that are both non-negative and strictly positive definite include the Gaussian kernel with bandwidth $h > 0$ defined for any $z, z' \in \mathcal{Z}$ as

$$K(z, z') = \exp(- \|z - z'\|_2^2 / h^2),$$

the Laplacian kernel, the Cauchy kernel, the Matérn kernel, the rational quadratic kernel, etc.

It is well-known that the non-parametric statistical estimator obtained by solving (1) is sensitive to the corruptions of the training data [9, 21, 28]. Similar phenomenon is also observed in machine learning where the solution of the risk minimization problem (1) is not guaranteed to be robust or generalizable [1, 2, 12, 14, 19, 23, 41, 42, 43]. The quality of the solution to (1) also deteriorates if the training sample size $N$ is small. Reweighting, obtained by modifying $\omega(\widehat{z}_i)$, is arising as an attractive resolution to improve robustness and enhance the out-of-sample performance in the test data [30, 34, 40]. At the same time, reweighting schemes have shown to produce many favorable effects: reweighting can increase fairness [15, 20, 38], and can also effectively handle covariate shift [10, 17, 44].

While reweighting has been successfully applied to the *empirical* risk minimization regime in which the weights are uniformly $1/N$, reweighting the samples when the weighting function $\omega$ is tied to a kernel is not a trivial task. In fact, the kernel captures inherently the *relative* positions of the relevant covariates $\widehat{z}$, and any reweighting scheme should also reflect these relationship in a global viewpoint. Another difficulty also arises due to the lack of convexity or concavity, which prohibits the modifications of the kernel parameters. For example, the mapping $h \mapsto \exp(-\|z - z'\|_2^2/h^2)$ for the Gaussian kernel is neither convex nor concave if $z \neq z'$. Thus, it is highly challenging to optimize over $h$ in the bandwidth parameter space. Alternatively, modifying the covariates $(\widehat{z}_i)_{i=1}^N$ will also result in reweighting effects. Nevertheless, optimizing over the covariates is intractable for sophisticated kernels such as the Matérn kernel.

**Contributions.** This paper relies fundamentally on an observation that the Gram matrix of a non-negative, (strictly) positive definite kernel is a non-negative, positive (semi)definite (also known as *doubly non-negative*) matrix. It is thus natural to modify the weights by modifying the corresponding matrix parametrization in an appropriate manner. Our contributions in this paper are two-fold:

- We propose a novel scheme for reweighting using a reparametrization of the sample weights as a doubly non-negative matrix. The estimate is characterized as the solution to a min-max optimization problem, in which the admissible values of the weights are obtained through a projection of an uncertainty set from the matrix space.

- We report in-depth analysis on two reweighting approaches based on the construction of the matrix uncertainty set with the log-determinant divergence and the Bures-Wasserstein distance. Exploiting strong duality, we show that the worst-case loss function and its gradient can be efficiently evaluated by solving the *univariate* dual problems. Consequently, the adversarially reweighted estimate can be found efficiently using first-order methods.

**Organization of the paper.** Section 2 introduces our generic framework of reweighting using doubly non-negative matrices. Sections 3 and 4 study two distinctive ways to customize our reweighting framework using the log-determinant divergence and the Bures-Wasserstein distance. Section 5 empirically illustrates that our reweighting strategy delivers promising results in the conditional expectation task based on numerous real life datasets. We have released code for these proposed tools[2].

**Notations.** The identity matrix is denoted by $I$. For any $A \in \mathbb{R}^{p \times p}$, $\operatorname{Tr}[A]$ denotes the trace of $A$, $A \geq 0$ means that all entries of $A$ are nonnegative. Let $\mathbb{S}^p$ denote the vector space of $p$-by-$p$ real and symmetric matrices. The set of positive (semi-)definite matrices is denoted by $\mathbb{S}_{++}^p$ (respectively, $\mathbb{S}_+^p$). For any $A, B \in \mathbb{R}^{p \times p}$, we use $\langle A, B \rangle = \operatorname{Tr}[A^\top B]$ to denote the Frobenius inner product between $A$ and $B$, and $\|v\|_2$ to denote the Euclidean norm of $v \in \mathbb{R}^p$.

## 2 A Reweighting Framework with Doubly Non-negative Matrices

We delineate in this section our reweighting framework using doubly non-negative matrices. This framework relies on the following observation: we can *reparametrize* the weights in (1) into a matrix $\widehat{\Omega}$ and the loss terms in (1) into a matrix $V(\beta)$, and the solution to the estimation problem (1) can be equivalently characterized as the minimizer of the problem

$$\min_{\beta} \ \langle \widehat{\Omega}, V(\beta) \rangle. \tag{3}$$

Notice that there may exist multiple equivalent reparametrizations of the form (3). However, in this paper, we focus on one specific parametrization where $\widehat{\Omega}$ is the nominal matrix of weights

$$\widehat{\Omega} = \begin{bmatrix} \widehat{\Omega}_{00} & \widehat{\Omega}_{01} & \cdots & \widehat{\Omega}_{0N} \\ \widehat{\Omega}_{10} & \widehat{\Omega}_{11} & \cdots & \widehat{\Omega}_{1N} \\ \vdots & \vdots & \ddots & \vdots \\ \widehat{\Omega}_{N0} & \widehat{\Omega}_{N1} & \cdots & \widehat{\Omega}_{NN} \end{bmatrix} \in \mathbb{S}^{N+1}$$

---

[2] https://github.com/lttam/Adversarial-Regression

with the elements being given by the weighting function $\omega$ as $\widehat{\Omega}_{0i} = \widehat{\Omega}_{i0} = \omega(\widehat{z}_i)$ for $i = 1, \ldots, N$, and the matrix-valued mapping $V : \beta \mapsto V(\beta) \in \mathbb{S}^{N+1}$ satisfies

$$V(\beta) = \begin{bmatrix} 0 & \ell(\beta, \widehat{x}_1, \widehat{y}_1) & \cdots & \ell(\beta, \widehat{x}_N, \widehat{y}_N) \\ \ell(\beta, \widehat{x}_1, \widehat{y}_1) & 0 & \cdots & 0 \\ \vdots & \vdots & \ddots & \vdots \\ \ell(\beta, \widehat{x}_N, \widehat{y}_N) & 0 & \cdots & 0 \end{bmatrix}.$$

A simple calculation reveals that the objective function of (3) is equivalent to that of (1) up to a positive constant factor of 2. As a consequence, their solutions coincide.

Problem (3) is an overparametrized reformulation of the weighted risk minimization problem (1). Indeed, the objective function of problem (3) involves an inner product of two symmetric matrices, while problem (1) can be potentially reformulated using an inner product of two vectors. While lifting the problem to the matrix space is not necessarily the most efficient approach, it endows us with more flexibility to perturb the weights in a coherent manner. This flexibility comes from the following two observations: (i) there may exist multiple matrices that can be used as the nominal matrix $\widehat{\Omega}$, and one can potentially choose $\widehat{\Omega}$ to improve the quality of the estimator, (ii) the geometry of the space of positive (semi)definite matrices is richer than the space of vectors.

To proceed, we need to make the following assumption.

**Assumption 2.1** (Regularity conditions). *The following assumptions hold throughout the paper.*

 *(i)  The function $\ell$ is nonnegative, and $\ell(\,\cdot\,, x, y)$ is convex, continuously differentiable for any $(x, y)$.*

*(ii)  The nominal weighting matrix $\widehat{\Omega}$ is symmetric positive definite and nonnegative.*

In this paper, we propose to find an estimate $\beta^\star$ that solves the following adversarially reweighted estimation problem

$$\min_\beta \max_{\Omega \in \mathcal{U}_{\varphi,\rho}(\widehat{\Omega})} \big\langle \Omega, V(\beta) \big\rangle \tag{4}$$

for some set $\mathcal{U}_{\varphi,\rho}(\widehat{\Omega})$ of feasible weighting matrices. The estimate $\beta^\star$ thus minimizes the worst-case loss uniformly over all possible perturbations of the weight $\Omega \in \mathcal{U}_{\varphi,\rho}(\widehat{\Omega})$. In particular, we explore the construction of the uncertainty set $\mathcal{U}_{\varphi,\rho}(\widehat{\Omega})$ that is motivated by the Gram matrix obtained via some non-negative and positive definite kernels. In this way, the weighting matrix can capture more information on the pair-wise relation among training data. Hence, it is reasonable to consider the set $\mathcal{U}_{\varphi,\rho}(\widehat{\Omega})$ of the form

$$\mathcal{U}_{\varphi,\rho}(\widehat{\Omega}) \triangleq \Big\{ \Omega \in \mathbb{S}_+^{N+1} : \Omega \geq 0, \; \varphi\big(\Omega, \widehat{\Omega}\big) \leq \rho \Big\}. \tag{5}$$

By definition, any $\Omega \in \mathcal{U}_{\varphi,\rho}(\widehat{\Omega})$ is a symmetric, positive semidefinite matrix and all elements of $\Omega$ are nonnegative. A matrix with these properties is called *doubly nonnegative*. From a high level perspective, the set $\mathcal{U}_{\varphi,\rho}(\widehat{\Omega})$ is defined as a ball of radius $\rho$ centered at the nominal matrix $\widehat{\Omega}$ and this ball is prescribed by a pre-determined measure of dissimilarity $\varphi$. Throughout this paper, we prescribe the uncertainty set $\mathcal{U}_{\varphi,\rho}(\widehat{\Omega})$ using some divergence $\varphi$ on the space of symmetric, positive semidefinite matrices $\mathbb{S}_+^{N+1}$.

**Definition 2.2** (Divergence). *For any $N \in \mathbb{N}$, $\varphi$ is a divergence on the symmetric positive semidefinite matrix space $\mathbb{S}_+^{N+1}$ if it is: (i) **non-negative:** $\varphi(\Omega_1, \Omega_2) \geq 0$ for all $\Omega_1, \; \Omega_2 \in \mathbb{S}_+^{N+1}$, and (ii) **indiscernable:** if $\varphi(\Omega_1, \Omega_2) = 0$ then $\Omega_1 = \Omega_2$.*

If we denote the adversarially reweighted loss function associated with $\mathcal{U}_{\varphi,\rho}(\widehat{\Omega})$ by

$$F_{\varphi,\rho}(\beta) \triangleq \max_{\Omega \in \mathcal{U}_{\varphi,\rho}(\widehat{\Omega})} \big\langle \Omega, V(\beta) \big\rangle,$$

then $\beta^\star$ can be equivalently rewritten as

$$\beta^\star = \arg\min_\beta F_{\varphi,\rho}(\beta). \tag{6}$$

A direct consequence is that the function $F_{\varphi,\rho}$ is convex in $\beta$ as long as the loss function $\ell$ satisfies the convex property of Assumption 2.1(i). Hence, the estimate $\beta^\star$ can be found efficiently using convex optimization provided that the function $F_{\varphi,\rho}$ and its gradient can be efficiently evaluated. Moreover, because $\varphi$ is a divergence, $\mathcal{U}_{\varphi,0}(\widehat{\Omega}) = \{\widehat{\Omega}\}$. Hence by setting $\rho = 0$, we will recover the nominal estimate that solves (1). In Section 3 and 4, we will subsequently specify two possible choices of $\varphi$ that lead to the desired efficiency in computing $F_{\varphi,\rho}$ as well as its gradient. Further discussion on Assumption 2.1 is relegated to the appendix. We close this section by discussing the robustness effects of our weighting scheme (4) on the conditional expectation estimation problem.

**Remark 2.3** (Connection to distributionally robust optimization)**.** *Consider the conditional expectation estimation setting, in which $\mathbb{E}[Y|Z = z_0]$ is the solution of the minimum mean square error estimation problem*

$$\mathbb{E}[Y|Z = z_0] = \arg\min_{\beta} \ \mathbb{E}[(\beta - Y)^2 | Z = z_0].$$

*In this setting, our reweighting scheme* (4) *coincides with the following distributionally robust optimization problem*

$$\min_{\beta} \ \max_{\mathbb{Q}_{Y|Z=z_0} \in \mathcal{B}(\hat{\mathbb{P}}_{Y|Z=z_0})} \ \mathbb{E}_{\mathbb{Q}_{Y|Z=z_0}}[(\beta - Y)^2],$$

*with the nominal conditional distribution defined as $\hat{\mathbb{P}}_{Y|Z=z_0}(\mathrm{d}y) \propto \sum_{i=1}^{N} K(z_0, \hat{z}_i)\delta_{\hat{y}_i}(\mathrm{d}y)$. The ambiguity set $\mathcal{B}(\hat{\mathbb{P}}_{Y|Z=z_0})$ is a set of conditional probability measures of $Y|Z = z_0$ constructed specifically as*

$$\mathcal{B}(\hat{\mathbb{P}}_{Y|Z=z_0}) = \left\{ \mathbb{Q}_{Y|Z=z_0} \ : \ \begin{array}{l} \exists \Omega \in \mathcal{U}_{\varphi,\rho}(\hat{\Omega}) \text{ so that } \Omega_{0i} = \Omega_{i0} = \omega(\hat{z}_i) \ \forall i \\ \mathbb{Q}_{Y|Z=z_0}(\mathrm{d}y) \propto \sum_{i=1}^{N} \omega(\hat{z}_i)\delta_{\hat{y}_i}(\mathrm{d}y) \end{array} \right\}.$$

Remark 2.3 reveals that our reweighting scheme recovers a specific robustification with distributional ambiguity. This robustification relies on using a kernel density estimate to construct the nominal conditional distribution, and the weights of the samples are induced by $\mathcal{U}_{\varphi,\rho}(\widehat{\Omega})$. Hence, our scheme is applicable for the emerging stream of robustifying conditional decisions, see [11, 18, 25, 26].

**Remark 2.4** (Choice of the nominal matrix)**.** *The performance of the estimate may depend on the specific choice of the nominal matrix $\widehat{\Omega}$. However, in this paper, we do not study this dependence in details. When the weights $\omega(\hat{z}_i)$ are given by a kernel, it is advised to choose $\widehat{\Omega}$ as the Gram matrix.*

## 3  Adversarial Reweighting Scheme using the Log-Determinant Divergence

We here study the adversarially reweighting scheme when the $\varphi$ is the log-determinant divergence.

**Definition 3.1** (Log-determinant divergence)**.** *For any positive integer $p \in \mathbb{N}$, the log-determinant divergence from $\Omega_1 \in \mathbb{S}_{++}^{p}$ to $\Omega_2 \in \mathbb{S}_{++}^{p}$ amounts to*

$$\mathbb{D}(\Omega_1, \Omega_2) \triangleq \mathrm{Tr}\left[\Omega_1 \Omega_2^{-1}\right] - \log\det(\Omega_1 \Omega_2^{-1}) - p.$$

The divergence $\mathbb{D}$ is the special instance of the log-determinant $\alpha$-divergence with $\alpha = 1$ [8]. Being a divergence, $\mathbb{D}$ is non-negative and it vanishes to zero if and only if $\Omega_1 = \Omega_2$. It is important to notice that the divergence $\mathbb{D}$ is only well-defined when both $\Omega_1$ and $\Omega_2$ are positive definite. Moreover, $\mathbb{D}$ is non-symmetric and $\mathbb{D}(\Omega_1, \Omega_2) \neq \mathbb{D}(\Omega_2, \Omega_1)$ in general. The divergence $\mathbb{D}$ is also tightly connected to the Kullback-Leibler divergence between two Gaussian distributions, and that $\mathbb{D}(\Omega_1, \Omega_2) = \mathrm{KL}(\mathcal{N}(0, \Omega_1) \| \mathcal{N}(0, \Omega_2))$, where $\mathcal{N}(0, \Omega)$ is a normal distribution with mean 0 and covariance matrix $\Omega$.

Suppose that $\widehat{\Omega}$ is invertible. Define the uncertainty set

$$\mathcal{U}_{\mathbb{D},\rho}(\widehat{\Omega}) = \{\Omega \in \mathbb{S}_{+}^{N+1} : \Omega \geq 0, \ \mathbb{D}(\Omega, \widehat{\Omega}) \leq \rho\}.$$

For any positive definite matrix $\widehat{\Omega}$, the function $\mathbb{D}(\,\cdot\,, \widehat{\Omega})$ is convex, thus the set $\mathcal{U}_{\mathbb{D},\rho}(\widehat{\Omega})$ is also convex. For this section, we examine the following optimal value function

$$F_{\mathbb{D},\rho}(\beta) = \max_{\Omega \in \mathcal{U}_{\mathbb{D},\rho}(\widehat{\Omega})} \ \langle \Omega, V(\beta) \rangle, \tag{7}$$

which corresponds to the worst-case reweighted loss using the divergence $\mathbb{D}$. The maximization problem (7) constitutes a nonlinear, convex semidefinite program. Leveraging a strong duality argument, the next theorem asserts that the complexity of evaluating $F_{\mathbb{D},\rho}(\beta)$ is equivalent to the complexity of solving a *univariate* convex optimization problem.

**Theorem 3.2** (Primal representation). *For any $\widehat{\Omega} \in \mathbb{S}_{++}^{N+1}$ and $\rho \in (0, +\infty)$, the function $F_{\mathbb{D},\rho}$ is convex. Moreover, for any $\beta$ such that $V(\beta) \neq 0$, let $\gamma^\star$ be the unique solution of the convex univariate optimization problem*

$$\inf_{\gamma I \succ \widehat{\Omega}^{\frac{1}{2}} V(\beta) \widehat{\Omega}^{\frac{1}{2}}} \gamma\rho - \gamma \log \det(I - \gamma^{-1} \widehat{\Omega}^{\frac{1}{2}} V(\beta) \widehat{\Omega}^{\frac{1}{2}}), \tag{8}$$

*then $F_{\mathbb{D},\rho}(\beta) = \langle \Omega^\star, V(\beta) \rangle$, where $\Omega^\star = \widehat{\Omega}^{\frac{1}{2}} [I - (\gamma^\star)^{-1} \widehat{\Omega}^{\frac{1}{2}} V(\beta) \widehat{\Omega}^{\frac{1}{2}}]^{-1} \widehat{\Omega}^{\frac{1}{2}}$. Moreover, the symmetric matrix $\Omega^\star$ is unique and doubly nonnegative.*

Notice that the condition $V(\beta) \neq 0$ is not restrictive: if $V(\beta) = 0$, then Assumption 2.1(i) implies that the incumbent solution $\beta$ incurs zero loss with $\ell(\beta, \widehat{x}_i, \widehat{y}_i) = 0$ for all $i$. In this case, $\beta$ is optimal and reweighting will produce no effect whatsoever. Intuitively, the infimum problem (8) is the dual counterpart of the supremum problem (7). The objective function of (8) is convex in the dual variable $\gamma$, and thus problem (8) can be efficiently solved using a gradient descent algorithm.

The gradient of $F_{\mathbb{D},\rho}$ is also easy to compute, as asserted in the following lemma.

**Lemma 3.3** (Gradient of $F_{\mathbb{D},\rho}$). *The function $F_{\mathbb{D},\rho}$ is continuously differentiable at $\beta$ with*

$$\nabla_\beta F_{\mathbb{D},\rho}(\beta) = 2 \sum_{i=1}^{N} \Omega_{0i}^\star \nabla_\beta \ell(\beta, \widehat{x}_i, \widehat{y}_i),$$

*where $\Omega^\star$ is defined as in Theorem 3.2 using the parametrization*

$$\Omega^\star = \begin{bmatrix} \Omega_{00}^\star & \Omega_{01}^\star & \cdots & \Omega_{0N}^\star \\ \Omega_{10}^\star & \Omega_{11}^\star & \cdots & \Omega_{1N}^\star \\ \vdots & \vdots & \ddots & \vdots \\ \Omega_{N0}^\star & \Omega_{N1}^\star & \cdots & \Omega_{NN}^\star \end{bmatrix}. \tag{9}$$

The proof of Lemma 3.3 exploits Danskin's theorem and the fact that $\Omega^\star$ is unique in Theorem 3.2. Minimizing $F_{\mathbb{D},\rho}$ is now achievable by applying state-of-the-art first-order methods.

**Sketch of Proof of Theorem 3.2.** The difficulty in deriving the dual formulation (8) lies in the non-negativity constraint $\Omega \geq 0$. In fact, this constraint imposes $(N + 1)(N + 2)/2$ individual component-wise constraints, and as such, simply dualizing problem (7) using a Lagrangian multiplier will entail a large number of auxiliary variables. To overcome this difficulty, we consider the relaxed set $\mathcal{V}_{\mathbb{D},\rho}(\widehat{\Omega}) \triangleq \{\Omega \in \mathbb{S}_+^{N+1} : \mathbb{D}(\Omega, \widehat{\Omega}) \leq \rho\}$. By definition, we have $\mathcal{U}_{\mathbb{D},\rho}(\widehat{\Omega}) \subseteq \mathcal{V}_{\mathbb{D},\rho}(\widehat{\Omega})$, and $\mathcal{V}_{\mathbb{D},\rho}(\widehat{\Omega})$ omits the nonnegativity requirement $\Omega \geq 0$. The set $\mathcal{V}_{\mathbb{D},\rho}(\widehat{\Omega})$ is also more amenable to optimization thanks to the following proposition.

**Proposition 3.4** (Properties of $\mathcal{V}_{\mathbb{D},\rho}(\widehat{\Omega})$). *For any $\widehat{\Omega} \in \mathbb{S}_{++}^{N+1}$ and $\rho \geq 0$, the set $\mathcal{V}_{\mathbb{D},\rho}(\widehat{\Omega})$ is convex and compact. Moreover, the support function of $\mathcal{V}_{\mathbb{D},\rho}(\widehat{\Omega})$ satisfies*

$$\delta_{\mathcal{V}_{\mathbb{D},\rho}(\widehat{\Omega})}^*(T) \triangleq \sup_{\Omega \in \mathcal{V}_{\mathbb{D},\rho}(\widehat{\Omega})} \text{Tr}[\Omega T] = \inf_{\substack{\gamma > 0 \\ \gamma \widehat{\Omega}^{-1} \succ T}} \gamma\rho - \gamma \log \det(I - \widehat{\Omega}^{\frac{1}{2}} T \widehat{\Omega}^{\frac{1}{2}}/\gamma)$$

*for any symmetric matrix $T \in \mathbb{S}^{N+1}$.*

Moreover, we need the following lemma which asserts some useful properties of the matrix $V(\beta)$.

**Lemma 3.5** (Properties of $V(\beta)$). *For any $\beta$, the matrix $V(\beta)$ is symmetric, nonnegative, and it has only two non-zero eigenvalues of value $\pm\sqrt{\sum_{i=1}^{N} \ell(\beta, \widehat{x}_i, \widehat{y}_i)^2}$.*

The proof of Theorem 3.2 proceeds by first constructing a tight upper bound for $F_{\mathbb{D},\rho}(\beta)$ as

$$F_{\mathbb{D},\rho}(\beta) \leq \max_{\Omega \in \mathcal{V}_{\mathbb{D},\rho}(\widehat{\Omega})} \langle \Omega, V(\beta) \rangle = \inf_{\gamma \widehat{\Omega}^{-1} \succ V(\beta)} \gamma\rho - \gamma \log \det(I - \frac{1}{\gamma} \widehat{\Omega}^{\frac{1}{2}} V(\beta) \widehat{\Omega}^{\frac{1}{2}}), \tag{10}$$

where the inequality in (10) follows from the fact that $\mathcal{U}_{\mathbb{D},\rho}(\widehat{\Omega}) \subseteq \mathcal{V}_{\mathbb{D},\rho}(\widehat{\Omega})$, and the equality follows from Proposition 3.4. Notice that $V(\beta)$ has one nonnegative eigenvalue by virtue of Lemma 3.5 and thus the constraint $\gamma\widehat{\Omega}^{-1} \succ V(\beta)$ already implies the condition $\gamma > 0$. Next, we argue that the optimizer $\Omega^\star$ of problem (10) can be constructed from the optimizer $\gamma^\star$ of the infimum problem via

$$\Omega^\star = \widehat{\Omega}^{\frac{1}{2}}[I - (\gamma^\star)^{-1}\widehat{\Omega}^{\frac{1}{2}}V(\beta)\widehat{\Omega}^{\frac{1}{2}}]^{-1}\widehat{\Omega}^{\frac{1}{2}}.$$

The last step involves proving that $\Omega^\star$ is a nonnegative matrix, and hence $\Omega^\star \in \mathcal{U}_{\mathbb{D},\rho}(\widehat{\Omega})$. As a consequence, the inequality (10) holds as an equality, which leads to the postulated result. The proof is relegated to the Appendix.

# 4 Adversarial Reweighting Scheme using the Bures-Wasserstein Type Divergence

In this section, we explore the construction of the set of possible weighting matrices using the Bures-Wasserstein distance on the space of positive semidefinite matrices.

**Definition 4.1** (Bures-Wasserstein divergence). *For any positive integer $p \in \mathbb{N}$, the Bures-Wasserstein divergence between $\Omega_1 \in \mathbb{S}_+^p$ and $\Omega_2 \in \mathbb{S}_+^p$ amounts to*

$$\mathbb{W}(\Omega_1, \Omega_2) \triangleq \mathrm{Tr}\left[\Omega_1 + \Omega_2 - 2\big(\Omega_2^{\frac{1}{2}}\Omega_1\Omega_2^{\frac{1}{2}}\big)^{\frac{1}{2}}\right].$$

For any positive semidefinite matrices $\Omega_1$ and $\Omega_2$, the value $\mathbb{W}(\Omega_1, \Omega_2)$ is equal to the square of the type-2 Wasserstein distance between two Gaussian distributions $\mathcal{N}(0, \Omega_1)$ and $\mathcal{N}(0, \Omega_2)$ [13]. As a consequence, $\mathbb{W}$ is a divergence: it is non-negative and indiscernable. However, $\mathbb{W}$ is not a proper distance because it may violate the triangle inequality. Compared to the divergence $\mathbb{D}$ studied in Section 3, the divergence $\mathbb{W}$ has several advantages as it is symmetric and is well-defined for all positive *semi*definite matrices. This divergence has also been of interest in quantum information, statistics, and the theory of optimal transport.

Given the nominal weighting matrix $\widehat{\Omega}$, we define the set of possible weighting matrices using the Bures-Wasserstein divergence $\mathbb{W}$ as

$$\mathcal{U}_{\mathbb{W},\rho}(\widehat{\Omega}) \triangleq \{\Omega \in \mathbb{S}_+^{N+1} : \Omega \geq 0,\ \mathbb{W}(\Omega, \widehat{\Omega}) \leq \rho\}.$$

Correspondingly, the worst-case loss function is

$$F_{\mathbb{W},\rho}(\beta) \triangleq \max_{\Omega \in \mathcal{U}_{\mathbb{W},\rho}(\widehat{\Omega})}\ \langle \Omega, V(\beta) \rangle. \tag{11}$$

**Theorem 4.2** (Primal representation). *For any $\widehat{\Omega} \in \mathbb{S}_+^{N+1}$ and $\rho \in (0, +\infty)$, the function $F_{\mathbb{W},\rho}$ is convex. Moreover, for any $\beta$ such that $V(\beta) \neq 0$, let $\gamma^\star$ be the unique solution of the convex univariate optimization problem*

$$\inf_{\gamma I \succ V(\beta)}\ \gamma(\rho - \mathrm{Tr}\,[\widehat{\Omega}]) + \gamma^2\langle(\gamma I - V(\beta))^{-1}, \widehat{\Omega}\rangle, \tag{12}$$

*then $F_{\mathbb{W},\rho}(\beta) = \langle\Omega^\star, V(\beta)\rangle$, where $\Omega^\star = (\gamma^\star)^2[\gamma^\star I - V(\beta)]^{-1}\widehat{\Omega}[\gamma^\star I - V(\beta)]^{-1}$. Moreover, the symmetric matrix $\Omega^\star$ is unique and doubly nonnegative.*

Thanks to the uniqueness of $\Omega^\star$ and Danskin's theorem, the gradient of $F_{\mathbb{W},\rho}$ is now a by-product of Theorem 4.2.

**Lemma 4.3** (Gradient of $F_{\mathbb{W},\rho}$). *The function $F_{\mathbb{W},\rho}$ is continuously differentiable at $\beta$ with*

$$\nabla_\beta F_{\mathbb{W},\rho}(\beta) = 2\sum_{i=1}^{N}\Omega_{0i}^\star\nabla_\beta\ell(\beta, \widehat{x}_i, \widehat{y}_i),$$

*where $\Omega^\star$ is defined as in Theorem 4.2 using the similar parametrization* (9).

A first-order minimization algorithm can be used to find the robust estimate with respect to the loss function $F_{\mathbb{W},\rho}$. Notice that problem (12) is one-dimensional, and either a bisection search or a gradient descent subroutine can be employed to solve (12) efficiently.

**Sketch of Proof of Theorem 4.2.** The proof of Theorem 4.2 follows a similar line of argument as the proof of Theorem 3.2. Consider the relaxed set $\mathcal{V}_{\mathbb{W},\rho}(\widehat{\Omega}) \triangleq \{\Omega \in \mathbb{S}_+^{N+1} : \mathbb{W}(\Omega, \widehat{\Omega}) \leq \rho\}$. By definition, $\mathcal{V}_{\mathbb{W},\rho}(\widehat{\Omega})$ omits the nonnegativity requirement $\Omega \geq 0$ and thus $\mathcal{U}_{\mathbb{W},\rho}(\widehat{\Omega}) \subseteq \mathcal{V}_{\mathbb{W},\rho}(\widehat{\Omega})$. The advantage of considering $\mathcal{V}_{\mathbb{W},\rho}(\widehat{\Omega})$ arises from the fact that the support function of the set $\mathcal{V}_{\mathbb{W},\rho}(\widehat{\Omega})$ admits a simple form [24, Proposition A.4].

**Proposition 4.4** (Properties of $\mathcal{V}_{\mathbb{W},\rho}(\widehat{\Omega})$). *For any $\widehat{\Omega} \in \mathbb{S}_+^{N+1}$ and $\rho \geq 0$, the set $\mathcal{V}_{\mathbb{W},\rho}(\widehat{\Omega})$ is convex and compact. Moreover, the support function of $\mathcal{V}_{\mathbb{W},\rho}(\widehat{\Omega})$ satisfies*

$$\delta_{\mathcal{V}_{\mathbb{D},\rho}(\widehat{\Omega})}^*(T) \triangleq \sup_{\Omega \in \mathcal{V}_{\mathbb{D},\rho}(\widehat{\Omega})} \mathrm{Tr}\left[\Omega T\right] = \inf_{\substack{\gamma > 0 \\ \gamma I \succ T}} \gamma(\rho - \mathrm{Tr}\left[\widehat{\Omega}\right]) + \gamma^2 \langle (\gamma I - T)^{-1}, \widehat{\Omega} \rangle.$$

The upper bound for $F_{\mathbb{W},\rho}(\beta)$ can be constructed as

$$F_{\mathbb{W},\rho}(\beta) \leq \max_{\Omega \in \mathcal{V}_{\mathbb{W},\rho}(\widehat{\Omega})} \langle \Omega, V(\beta) \rangle = \inf_{\gamma I \succ V(\beta)} \gamma(\rho - \mathrm{Tr}\left[\widehat{\Omega}\right]) + \gamma^2 \langle (\gamma I - V(\beta))^{-1}, \widehat{\Omega} \rangle, \qquad (13)$$

where the inequality in (13) follows from the fact that $\mathcal{U}_{\mathbb{W},\rho}(\widehat{\Omega}) \subseteq \mathcal{V}_{\mathbb{W},\rho}(\widehat{\Omega})$, and the equality follows from Proposition 4.4. In the second step, we argue that the optimizer $\Omega^\star$ of problem (13) can be constructed from the optimizer $\gamma^\star$ of the infimum problem via

$$\Omega^\star = (\gamma^\star)^2 [\gamma^\star I - V(\beta)]^{-1} \widehat{\Omega} [\gamma^\star I - V(\beta)]^{-1}.$$

The last step involves proving that $\Omega^\star$ is a nonnegative matrix by exploiting Lemma 3.5, and hence $\Omega^\star \in \mathcal{U}_{\mathbb{D},\rho}(\widehat{\Omega})$. Thus, inequality (13) is tight, leading to the desired result.

# 5 Numerical Experiments on Real Data

We evaluate our adversarial reweighting schemes on the conditional expectation estimation task. To this end, we use the proposed reweighted scheme on the NW estimator of Example 1.1. The robustification using the log-determinant divergence and the Bures-Wasserstein divergence are denoted by NW-LogDet and NW-BuresW, respectively. We compare our NW robust estimates against four popular baselines for estimating the conditional expectation: (i) the standard NW estimate in Example 1.1 with Gaussian kernel, (ii) the LLR estimate in Example 1.2 with Gaussian kernel[3], (iii) the intercepted $\beta_1$ of LLR estimate (i.e., only the first dimension of $\beta_{\mathrm{LLR}}$), denoted as LLR-I, and (iv) the NW-Metric [27] which utilizes the Mahalanobis distance in the Gaussian kernel.

**Datasets.** We use 8 real-world datasets: (i) abalone (`Abalone`), (ii) bank-32fh (`Bank`), (iii) cpu (`CPU`), (iv) kin40k (`KIN`), (v) elevators (`Elevators`), (vi) pol (`POL`), (vii) pumadyn32nm (`PUMA`), and (viii) slice (`Slice`) from the Delve datasets, the UCI datasets, the KEEL datasets and datasets in Noh et al. [27]. Due to space limitation, we report results on the first 4 datasets and relegate the remaining results to the Appendix. Datasets characteristics can also be found in the supplementary material.

**Setup.** For each dataset, we randomly split 1200 samples for training, 50 samples for validation to choose the bandwith $h$ of the Gaussian kernel, and 800 samples for test. More specially, we choose the squared bandwidth $h^2$ for the Gaussian kernel from a predefined set $\left\{10^{-2:1:4}, 2 \times 10^{-2:1:4}, 5 \times 10^{-2:1:4}\right\}$. For a tractable estimation, we follow the approach in Brundsdon et al. [7] and Silverman [35] to restrict the relevant samples to $N$ nearest neighbors of each test sample $z_i$ with $N \in \{10, 20, 30, 50\}$. The range of the radius $\rho$ has 4 different values $\rho \in \{0.01, 0.1, 1, 10\}$. Finally, the prediction error is measured by the root mean square error (RMSE), i.e., $\mathrm{RMSE} = \sqrt{n_t^{-1} \sum_{i=1}^{n_t} (\widehat{y}_i - \widehat{\beta}_i)^2}$, where $n_t$ is the test sample size (i.e., $n_t = 800$) and $\widehat{\beta}_i$ is the conditional expectation estimate at the test sample $z_i$. We repeat the above procedure 10 times to obtain the average RMSE. All our experiments are run on commodity hardware.

**Ideal case: no sample perturbation.** We first study how different estimators perform when there is no perturbation in the training data. In this experiment, we set the nearest neighbor size to $N = 50$, and our reweighted estimators are obtained with the uncertainty size of $\rho = 0.1$.

---

[3]We omit results of LLR in Figures 1 and 2 for a better visualization. See the Appendix for detailed results.

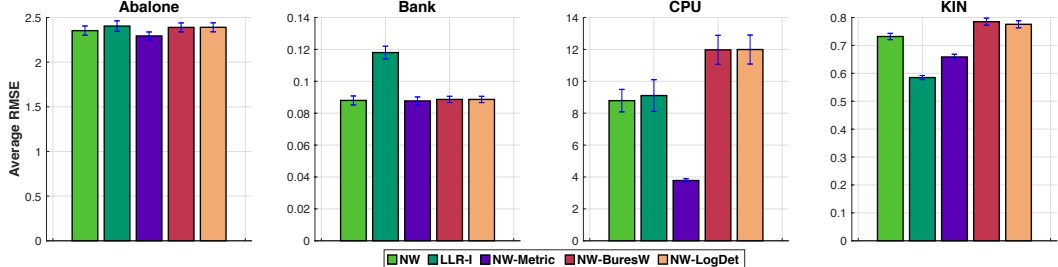

Figure 1: Average RMSE for ideal case with no perturbation.

Figure 1 shows the average RMSE across the datasets. The NW-Metric estimator outperforms the standard NW, which agrees with the empirical observation in Noh et al. [27]. More importantly, we observe that our adversarial reweighting schemes perform competitively against the baselines on several datasets.

**When training samples are perturbed.** We next evaluate the estimation performances when $\tau \in \{0.2N, 0.4N, 0.6N, 0.8N, N\}$ nearest samples from the $N$ training neighbors of each test sample are perturbed. We specifically generate perturbations only in the response dimension by shifting $y \mapsto \kappa y$, where $\kappa$ is sampled uniformly from $[1.8, 2.2]$. We set $N = 50$ and $\rho = 0.1$ as the experiment for the ideal case (no sample perturbation).

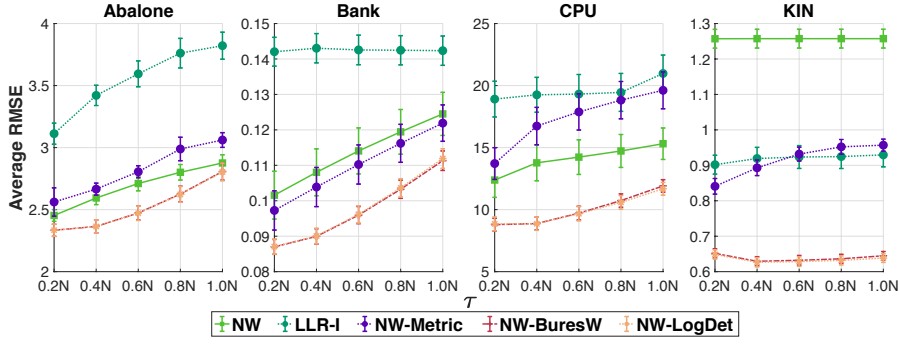

Figure 2: RMSE for varying perturbation levels $\tau$. Results are averaged over 10 independent replications, each contains 800 test samples.

Figure 2 shows the average RMSE for with varying perturbation level $\tau$. The performance of all baseline approaches severely deteriorate, while both NW-LogDet and NW-BuresW can alleviate the effect of data perturbation. Our adversarial reweighting schemes consistently outperform all baselines in all datasets for the perturbed training data, across all 5 perturbations $\tau$.

We then evaluate the effects of the uncertainty size $\rho$ and the nearest neighbor size $N$ on NW-LogDet and NW-BuresW.

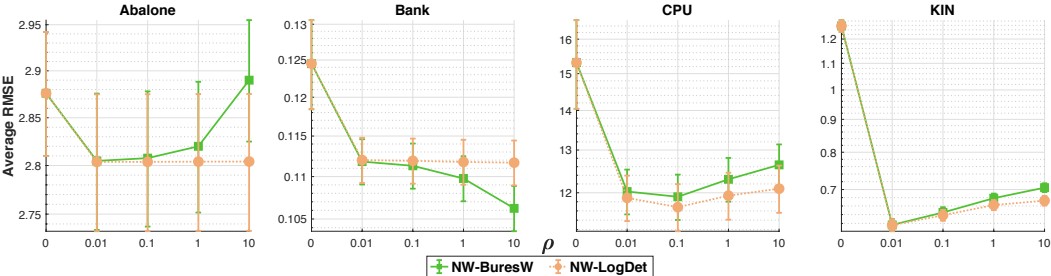

Figure 3: Average RMSE as a function of the ambiguity size $\rho$. Errors at $\rho = 0$ indicate the performance of the vanilla NW estimator.

**Effects of the uncertainty size $\rho$.** In this experiment, we set the nearest neighbor size to $N = 50$, the perturbation $\tau = N$. Figure 3 illustrates the effects of the uncertainty size $\rho$ for the adversarial reweighting schemes. Errors at $\rho = 0$ indicate the performance of the vanilla NW estimator. We observe that the adversarial reweighting schemes perform well at some certain $\rho$ and when that $\rho$ is increased more, the performances decrease. The uncertainty size $\rho$ plays an important role for the adversarial reweighting schemes in applications. Tuning $\rho$ may consequently improve the performances of the adversarial reweighting schemes.

**Effects of the nearest neighbor size $N$.** In this experiment, we set the uncertainty size to $\rho = 0.1$.

Figure 4 shows the effects of the nearest neighbor size $N$ for the adversarial reweighting schemes under varying perturbation $\tau$ in the KIN dataset. We observe that the performances of the adversarial reweighting schemes with $N \in \{20, 30\}$ perform better than those with $N \in \{10, 50\}$. Note that when $N$ is increased, the computational cost is also increased (see Equation (1) and Figure 6). Similar to the cases of the uncertainty size $\rho$, tuning $N$ may also help to improve performances of the adversarial reweighting schemes.

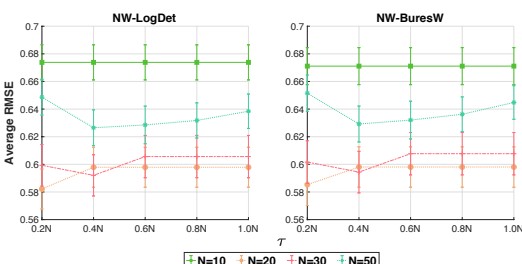

Figure 4: Effects of the nearest neighbor size $N$.

**Under varying shifting w.r.t. $\kappa$.** In this experiment, we set the nearest neighbor size to $N = 50$. Figure 5 illustrates the performances of NW-LogDet under varying shifting w.r.t. $\kappa$ in the KIN dataset. For the left plots of the Figures 5, we set the uncertainty size to $\rho = 0.1$ when varying the perturbation $\tau$. We observe that the adversarial reweighting schemes provide different degrees of mitigation for the perturbation under varying shifting w.r.t. $\kappa$. For the right plots of the Figures 5, we set the perturbation to $\tau = 0.2N$ when varying the uncertainty size $\rho$. We observe that the reweighting schemes under

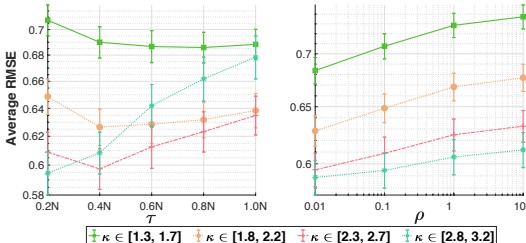

Figure 5: Effects of perturbation intensity $\kappa$ for NW-LogDet estimate. Left plot: different perturbation $\tau$, right plot: different uncertainty size $\rho$.

varying shifting w.r.t. $\kappa$ have the same behaviors as in Figure 3 when we consider the effects of the uncertainty size $\rho$ (i.e., when $\kappa \in [1.8, 2.2]$). Similar results for NW-BuresW are reported in the supplementary material.

**Time consumption.** In Figure 6, we illustrate the time consumption for the adversarial reweighting schemes under varying neighbor size $N$ and uncertainty size $\rho$ in the KIN dataset. The adversarial reweighting schemes averagely take about 10 seconds for their estimation. When $N$ and/or $\rho$ increases, the computation of the adversarial reweighting schemes take longer. This is intuitive because the dimension of the weight matrix $\Omega$ scales quadratically in

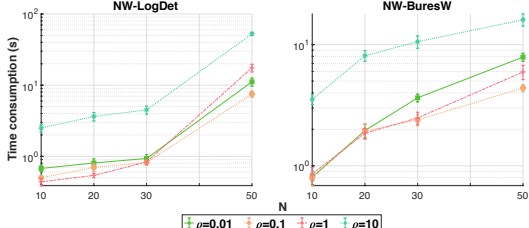

Figure 6: Effects of $N$ on computational time.

the neighbor size $N$. Bigger uncertainty size $\rho$ implies a larger feasible set $\mathcal{U}_{\varphi,\rho}(\widehat{\Omega})$, which leads to longer computing time to evaluate $F_{\varphi,\rho}$ and its gradient.

**Concluding Remarks.** We introduce two novel schemes for sample reweighting using matrix reparametrization. These two invariants are particularly attractive when the original weights are given through kernel functions. Note that the adversarial reweighting with Bures-Wasserstein distance $\mathbb{W}$ can be generalized to cases where the nominal weighting matrix $\widehat{\Omega}$ is singular, unlike the reweighting with the log-determinant divergence $\mathbb{D}$.

**Remark 5.1** (Invariant under permutation). *Our results hold under any simultaneous row and column permutation of the nominal weighting matrix $\widehat{\Omega}$ and the mapping $V(\beta)$. To see this, let $P$ be any $(N+1)$-dimensional permutation matrix, and let $\widehat{\Omega}_P \triangleq P\widehat{\Omega}P$ and $V_P(\beta) \triangleq PV(\beta)P$. Then*

$$\max_{\Omega \in \mathcal{U}_{\varphi,\rho}(\widehat{\Omega}_P)} \langle \Omega, V_P(\beta) \rangle = \langle P\Omega^\star P, V_P(\beta) \rangle = \langle \Omega^\star, V(\beta) \rangle,$$

*where $\Omega^\star$ is calculated as in Theorem 3.2 for $\varphi = \mathbb{D}$, and as in Theorem 4.2 for $\varphi = \mathbb{W}$. The proof relies on the fact that $P^\top P = PP^\top = I$, that both $\mathbb{D}$ and $\mathbb{W}$ are permutation invariant (in the sense that $\varphi(\Omega_1, \Omega_2) = \varphi(P\Omega_1 P, P\Omega_2 P)$), and that the inner product is also permutation invariant. Similar results hold for the gradient information, and hence the optimal solution of $\beta$ is preserved under row and column permutations of $\widehat{\Omega}$ and $V(\beta)$.*

**Acknowledgements.** Material in this paper is based upon work supported by the Air Force Office of Scientific Research under award number FA9550-20-1-0397. Support is gratefully acknowledged from NSF grants 1915967, 1820942, 1838676, Simons Foundation grant (#318995), JSPS KAKENHI grant 20K19873, and MEXT KAKENHI grants 20H04243, 21H04874. We thank the anonymous referees for their constructive feedbacks that helped improve and clarify this paper.

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
