# *Supplementary Material for:* Adversarial Regression with Doubly Non-negative Weighting Matrices

In this supplementary, we give details for the proofs of our technical results in Section A. In Section B, we describe implementation details. In Section C, we illustrate additional empirical results.

We have released code for our proposed tools at:

https://github.com/lttam/Adversarial-Regression

## A   Proofs and Additional Discussions

### A.1   Proofs of Section 3

In the following, the symbol $\langle \cdot, \cdot \rangle$ will be used to represent both Frobenius norm of matrices and standard Euclidean norm of vectors. Also for a $p \times p$ matrix $A$, we let $\|A\|_2 = \sqrt{\langle A, A \rangle}$ to denote its Frobenius norm and let $\lambda_{\max}(A)$ to represent its maximum eigenvalue. In order to prove Proposition 3.4, we begin by computing the support function of the convex cone of symmetric positive definite (SPD) matrices.

**Lemma A.1** (Support function). *For any matrices $A \in \mathbb{S}^{N+1}$ and $\Omega \in \mathbb{S}^{N+1}_{++}$, we have $\mathrm{Tr}\left[A\Omega\right] \leq \lambda_{\max}(A)\,\mathrm{Tr}\left[\Omega\right]$, and $\mathrm{Tr}\left[A\Omega\right] < 0$ if in addition $A \preceq 0$ and $A \neq 0$. Also for each $A \in \mathbb{S}^{N+1}$,*

$$\sup_{\Omega \succ 0}\ \mathrm{Tr}\left[A\Omega\right] = \begin{cases} +\infty & \text{if } \lambda_{\max}(A) > 0, \\ 0 & \text{if } \lambda_{\max}(A) \leq 0. \end{cases}$$

*Proof of Lemma A.1.* Since $A$ is symmetric, we can decompose it as $D = Q^\top A Q$ with $D$ being a diagonal matrix formed by eigenvalues of $A$ and $Q$ being an orthogonal matrix whose columns are normalized eigenvectors of $A$. Then $Q^\top A = D Q^\top$ and hence

$$\mathrm{Tr}\left[A\Omega\right] = \mathrm{Tr}\left[Q^\top A \Omega Q\right] = \mathrm{Tr}\left[D Q^\top \Omega Q\right] \leq \lambda_{\max}(D)\,\mathrm{Tr}\left[Q^\top \Omega Q\right] = \lambda_{\max}(A)\,\mathrm{Tr}\left[\Omega\right],$$

where we have used the fact $Q^\top \Omega Q \succ 0$ to obtain the inequality. In case $A \preceq 0$ and $A \neq 0$, then as all diagonal elements of $D$ are nonpositive with at least one negative element we have $\mathrm{Tr}\left[A\Omega\right] = \mathrm{Tr}\left[D Q^\top \Omega Q\right] < 0$. These give the first part of the lemma.

For the second part, let $v$ be an eigenvector of $A$ corresponding to eigenvalue $\lambda_{\max}(A)$. In case $\lambda_{\max}(A) > 0$, by taking $\Omega_t = I + t v v^\top \succ 0$ with $t \in (0, +\infty)$ and letting $t \to +\infty$ we see that $\mathrm{Tr}\left[A\Omega_t\right] = \mathrm{Tr}\left[A\right] + t\lambda_{\max}(A)\,\mathrm{Tr}\left[vv^\top\right]$ tends to $+\infty$. This implies that $\sup_{\Omega \succ 0}\ \mathrm{Tr}\left[A\Omega\right] = +\infty$. For the case $\lambda_{\max}(A) \leq 0$, we instead take $\Omega_t = t I + t v v^\top \succ 0$ with $t \in (0, 1)$ and let $t \to 0^+$ to conclude that $\sup_{\Omega \succ 0}\ \mathrm{Tr}\left[A\Omega\right] \geq 0$. On the other hand, due to the first part of the lemma we always have $\sup_{\Omega \succ 0}\ \mathrm{Tr}\left[A\Omega\right] \leq 0$. Therefore, the desired result follows. □

**For Proposition 3.4**

*Proof of Proposition 3.4.* The set $\mathcal{V}_{\mathbb{D},\rho}(\widehat{\Omega})$ is bounded. Indeed, if $\rho = 0$ then $\mathcal{V}_{\mathbb{D},\rho}(\widehat{\Omega}) = \{\widehat{\Omega}\}$ which is trivially bounded. If $\rho > 0$, then any $\Omega \in \mathcal{V}_{\mathbb{D},\rho}(\widehat{\Omega})$ satisfies $\mathrm{Tr}\left[\Omega\widehat{\Omega}^{-1}\right] - \log\det(\Omega\widehat{\Omega}^{-1}) - (N +$

$1) \leq \rho$. Since $\Omega\widehat{\Omega}^{-1}$ is similar to the matrix $\widehat{\Omega}^{-\frac{1}{2}}\Omega\widehat{\Omega}^{-\frac{1}{2}}$, we can rewrite this as

$$\mathrm{Tr}\left[\widehat{\Omega}^{-\frac{1}{2}}\Omega\widehat{\Omega}^{-\frac{1}{2}}\right] - \log\det(\widehat{\Omega}^{-\frac{1}{2}}\Omega\widehat{\Omega}^{-\frac{1}{2}}) - (N+1) \leq \rho. \tag{A.1}$$

We claim that this implies that $\Omega$ is bounded in the sense that there exist numbers $0 < \underline{\sigma} \leq \bar{\sigma} < \infty$ depending only on $\rho$, $\widehat{\Omega}$, and $N$ such that $\underline{\sigma}I \preceq \Omega \preceq \bar{\sigma}I$. To see this, let $\sigma_i$ $(i = 1, ..., N+1)$ be the eigenvalues of $\widehat{\Omega}^{-\frac{1}{2}}\Omega\widehat{\Omega}^{-\frac{1}{2}}$. Then (A.1) gives

$$\sum_{i=1}^{N+1}[\sigma_i - \log\sigma_i - 1] \leq \rho.$$

Because the function $\sigma \mapsto \sigma - \log\sigma - 1$ is non-negative for every $\sigma > 0$, we then find that $\{\sigma_i - \log\sigma_i - 1\}_{i=1}^{N+1}$ is bounded. Therefore, $\sigma_i$ must be bounded between two positive constants depending only on $\rho$ and $N$. Now let $\lambda$ be an eigenvalue of $\Omega$ and let $v$ be its associated eigenvector. Then $\lambda\|v\|_2^2 = \langle v, \Omega v\rangle = \langle(\widehat{\Omega}^{\frac{1}{2}}v), \widehat{\Omega}^{-\frac{1}{2}}\Omega\widehat{\Omega}^{-\frac{1}{2}}(\widehat{\Omega}^{\frac{1}{2}}v)\rangle$, and so

$$\sigma_*\|\widehat{\Omega}^{\frac{1}{2}}v\|_2^2 \leq \lambda\|v\|_2^2 \leq \sigma^*\|\widehat{\Omega}^{\frac{1}{2}}v\|_2^2,$$

where $\sigma_*$ and $\sigma^*$ respectively denote the smallest and largest eigenvalues of $\widehat{\Omega}^{-\frac{1}{2}}\Omega\widehat{\Omega}^{-\frac{1}{2}}$. It follows that $\sigma_*\lambda_{\min}(\widehat{\Omega}) \leq \lambda \leq \sigma^*\lambda_{\max}(\widehat{\Omega})$. Thus all eigenvalues of $\Omega$ are bounded between two positive constants, and the claim is proved.

One now can rewrite $\mathcal{V}_{\mathbb{D},\rho}(\widehat{\Omega})$ as

$$\mathcal{V}_{\mathbb{D},\rho}(\widehat{\Omega}) = \left\{\Omega \in \mathbb{S}_+^{N+1} : \underline{\sigma}I \preceq \Omega \preceq \bar{\sigma}I, \ \mathrm{Tr}\left[\Omega\widehat{\Omega}^{-1}\right] - \log\det(\Omega\widehat{\Omega}^{-1}) - (N+1) \leq \rho\right\}.$$

The function $\Omega \mapsto \mathrm{Tr}\left[\Omega\widehat{\Omega}^{-1}\right] - \log\det(\Omega\widehat{\Omega}^{-1})$ is convex and continuous on the set $\{\Omega : \underline{\sigma}I \preceq \Omega \preceq \bar{\sigma}I\}$, thus the set $\mathcal{V}_{\mathbb{D},\rho}(\widehat{\Omega})$ is convex and compact.

We now proceed to provide the support function of $\mathcal{V}_{\mathbb{D},\rho}(\widehat{\Omega})$. One can verify that $\widehat{\Omega}$ is the Slater point of the convex set $\mathcal{V}_{\mathbb{D},\rho}(\widehat{\Omega})$. Assume momentarily that $T \neq 0$, using a duality argument, we find

$$\sup_{\Omega \in \mathcal{V}_{\mathbb{D},\rho}(\widehat{\Omega})} \mathrm{Tr}\left[T\Omega\right]$$

$$= \sup_{\Omega \succ 0}\inf_{\gamma \geq 0} \mathrm{Tr}\left[T\Omega\right] + \gamma\left(\bar{\rho} - \mathrm{Tr}\left[\widehat{\Omega}^{-1}\Omega\right] + \log\det\Omega\right)$$

$$= \inf_{\gamma \geq 0}\left\{\gamma\bar{\rho} + \sup_{\Omega \succ 0}\left\{\mathrm{Tr}\left[(T - \gamma\widehat{\Omega}^{-1})\Omega\right] + \gamma\log\det\Omega\right\}\right\}, \tag{A.2}$$

where the last equality follows from strong duality [1, Proposition 5.3.1], and $\bar{\rho} \triangleq \rho + (N+1) - \log\det\widehat{\Omega} \in \mathbb{R}$. Note that $\gamma = 0$ is not an optimal solution to the minimization problem (A.2). Indeed, if the maximum eigenvalue of $T$ is strictly positive, then the objective value of problem (A.2) evaluated at $\gamma = 0$ is $+\infty$ by Lemma A.1. However, because $\mathcal{V}_{\mathbb{D},\rho}(\widehat{\Omega})$ is compact and bounded, we have $\sup_{\Omega \in \mathcal{V}_{\mathbb{D},\rho}(\widehat{\Omega})} \mathrm{Tr}\left[T\Omega\right] < \infty$. In case the maximum eigenvalue of $T$ is nonpositive, then from Lemma A.1 we see that the objective value of problem (A.2) evaluated at $\gamma = 0$ is $0$, while $\sup_{\Omega \in \mathcal{V}_{\mathbb{D},\rho}(\widehat{\Omega})} \mathrm{Tr}\left[T\Omega\right] < 0$. Thus, the infimum in problem (A.2) can be restricted to $\gamma > 0$.

If $T - \gamma\widehat{\Omega}^{-1} \not\prec 0$, then the inner supremum problem in (A.2) becomes unbounded according to Lemma A.1. If $T - \gamma\widehat{\Omega}^{-1} \prec 0$ then the inner supremum problem admits the unique optimal solution

$$\Omega^\star(\gamma) = \gamma(\gamma\widehat{\Omega}^{-1} - T)^{-1}, \tag{A.3}$$

which is obtained by solving the first-order optimality condition. By placing this optimal solution into the objective function and arranging terms together with using the definition of $\bar{\rho}$, we have

$$\sup_{\Omega \in \mathcal{V}_{\mathbb{D},\rho}(\widehat{\Omega})} \mathrm{Tr}\left[T\Omega\right] = \inf_{\substack{\gamma > 0 \\ \gamma\widehat{\Omega}^{-1} \succ T}} \gamma\rho - \gamma\log\det(I - \gamma^{-1}\widehat{\Omega}^{\frac{1}{2}}T\widehat{\Omega}^{\frac{1}{2}}). \tag{A.4}$$

To complete the proof, we show that the reformulation (A.4) holds even for $T = 0$ or $\rho = 0$. Notice that if $T = 0$ or $\rho = 0$, then the left-hand side of (A.4) evaluates to $0$. If $T = 0$, the infimum problem

on the right-hand side of (A.4) also attains the optimal value of 0 asymptotically as $\gamma$ decreases to 0. If $\rho = 0$ and $T \neq 0$, then the infimum problem on the right-hand side of (A.4) also attains the optimal value of 0 asymptotically as $\gamma$ increases to $+\infty$ by the l'Hopital rule

$$\lim_{\gamma \uparrow +\infty} -\gamma \log \det(I - \widehat{\Omega}^{\frac{1}{2}} T \widehat{\Omega}^{\frac{1}{2}}/\gamma) = \mathrm{Tr}\left[T\widehat{\Omega}\right].$$

This observation completes the proof. $\qquad \square$

For an $p \times p$ real matrix $A$, its spectral radius $\mathcal{R}(A)$ is defined as the largest absolute value of its eigenvalues. The following elementary fact is well known.

**Lemma A.2** (Nonnegativity of inverse). *Let $A$ be an $p \times p$ real matrix such that $\mathcal{R}(A) < 1$ and all its entries are nonnegative. Then the matrix $I - A$ is invertible and all entries of $(I - A)^{-1}$ are nonnegative.*

*Proof of Lemma A.2.* For completeness, we include a proof here. By Gelfand's formula, we have

$$\lim_{k \to +\infty} \|A^k\|_2^{\frac{1}{k}} = \mathcal{R}(A) < 1.$$

Thus there exist constants $\tau \in (0, 1)$ and $k_0 \in \mathbb{N}$ such that $\|A^k\|_2 < \tau^k$ for every $k \geq k_0$. Therefore, the Neumann series

$$\sum_{k=0}^{\infty} A^k = (I - A)^{-1}$$

converges. This together with the assumption about the nonnegativity of $A$ implies that all entries of $(I - A)^{-1}$ are nonnegative. $\qquad \square$

**For Theorem 3.2**

*Proof of Theorem 3.2.* We find

$$F_{\mathbb{D},\rho}(\beta) = \max_{\Omega \in \mathcal{U}_{\mathbb{D},\rho}(\widehat{\Omega})} \left\langle \Omega, V(\beta) \right\rangle \tag{A.5a}$$

$$\leq \max_{\Omega \in \mathcal{V}_{\mathbb{D},\rho}(\widehat{\Omega})} \left\langle \Omega, V(\beta) \right\rangle \tag{A.5b}$$

$$= \inf_{\gamma I \succ \widehat{\Omega}^{\frac{1}{2}} V(\beta) \widehat{\Omega}^{\frac{1}{2}}} \gamma\rho - \gamma \log \det(I - \gamma^{-1}\widehat{\Omega}^{\frac{1}{2}} V(\beta) \widehat{\Omega}^{\frac{1}{2}}), \tag{A.5c}$$

where equality (A.5a) is the definition of $F_{\mathbb{D},\rho}$ as in (7), inequality (A.5b) follows from the fact that $\mathcal{U}_{\mathbb{D},\rho}(\widehat{\Omega}) \subseteq \mathcal{V}_{\mathbb{D},\rho}(\widehat{\Omega})$, and equality (A.5c) follows from Proposition 3.4. Notice that $V(\beta)$ has one nonnegative eigenvalue by virtue of Lemma 3.5 and thus the constraint $\gamma I \succ \widehat{\Omega}^{\frac{1}{2}} V(\beta) \widehat{\Omega}^{\frac{1}{2}}$ implies the condition $\gamma > 0$.

The strictly positive radius condition $\rho \in (0, +\infty)$ implies the existence of a Slater point $\widehat{\Omega}$ of the set $\mathcal{U}_{\mathbb{D},\rho}(\widehat{\Omega})$. By [1, Proposition 5.5.4], the existence of a solution $\gamma^\star > 0$ that minimizes the dual problem (A.5c) is guaranteed. Moreover, the objective function of problem (A.5c) is strictly convex, and thus the solution $\gamma^\star$ is unique. By inspecting (A.3), the solution

$$\Omega^\star(\gamma^\star) = (\widehat{\Omega}^{-1} - \frac{1}{\gamma^\star} V(\beta))^{-1}$$

thus solves the primal problem (A.5b) by [1, pp. 178].

Assumption 2.1 implies that both $\widehat{\Omega}$ and $V(\beta)/\gamma^\star$ are nonnegative matrices. Also the spectral radius of $(\gamma^\star)^{-1}\widehat{\Omega}^{\frac{1}{2}} V(\beta) \widehat{\Omega}^{\frac{1}{2}}$ is smaller than 1 by the feasibility of $\gamma^\star$ in problem (A.5c). Hence the matrix $[I - (\gamma^\star)^{-1}\widehat{\Omega}^{\frac{1}{2}} V(\beta) \widehat{\Omega}^{\frac{1}{2}}]^{-1}$ is nonnegative by Lemma A.2. As $\Omega^\star(\gamma^\star) = \widehat{\Omega}^{\frac{1}{2}}[I - (\gamma^\star)^{-1}\widehat{\Omega}^{\frac{1}{2}} V(\beta) \widehat{\Omega}^{\frac{1}{2}}]^{-1}\widehat{\Omega}^{\frac{1}{2}}$, we conclude that $\Omega^\star(\gamma^\star)$ is a matrix with nonnegative entries. Moreover, $\Omega^\star(\gamma^\star)$ is also positive semidefinite. Thus $\Omega^\star(\gamma^\star)$ is doubly nonnegative. This observation completes the proof. $\qquad \square$

**For Lemma 3.3**

*Proof of Lemma 3.3.* Because $\mathcal{U}_{\mathbb{D},\rho}(\widehat{\Omega})$ is compact, by [2, Proposition A.22], the subdifferential of $F_{\mathbb{D},\rho}$ is

$$\partial F_{\mathbb{D},\rho}(\beta) = \operatorname{ConvexHull}\Big(2\sum_{i=1}^{N} \Omega_{0i}^{\star}\nabla_{\beta}\ell(\beta,\widehat{x}_i,\widehat{y}_i) \,\Big|\, \Omega^{\star} \in \mathcal{O}^{\star}(\beta)\Big),$$

where $\mathcal{O}^{\star}(\beta) = \big\{\Omega \in \mathcal{U}_{\mathbb{D},\rho}(\widehat{\Omega}) : \langle \Omega, V(\beta)\rangle = F_{\mathbb{D},\rho}(\beta)\big\}$ is the optimal solution set. By Theorem 3.2, the optimal solution set $\mathcal{O}^{\star}(\beta)$ is a singleton, which leads to the postulated result. $\square$

**For Lemma 3.5**

*Proof of Lemma 3.5.* The symmetry of $V(\beta)$ follows from definition. The nonnegativity of $V(\beta)$ follows from Assumption 2.1. Let $\lambda$ be an eigenvalue of $V(\beta)$, then $\lambda$ solves the characteristic equation $\det(V(\beta) - \lambda I) = 0$. By exploiting the form of $V(\beta)$ and by the determinant formula for the arrowhead matrix, the eigenvalue $\lambda$ then solves the algebraic equation $\lambda^{N-1}\big[\lambda^2 - \sum_{i=1}^{N}\ell(\beta,\widehat{x}_i,\widehat{y}_i)^2\big] = 0$. This completes the proof. $\square$

## A.2   Proofs of Section 4

**For Theorem 4.2**

*Proof of Theorem 4.2.* The statement regarding $\gamma^{\star}$ and the expression of $F_{\mathbb{W},\rho}$ and $\Omega^{\star}$ follows from [3, Proposition A.4]. It remains to show that $\Omega^{\star}$ is nonnegative. Indeed, the constraint $\gamma^{\star}I \succ V(\beta)$ implies that the spectral radius of $(\gamma^{\star})^{-1}V(\beta)$ is smaller than 1. Therefore, the inverse matrix $[I - (\gamma^{\star})^{-1}V(\beta)]^{-1}$ is nonnegative according to Lemma A.2. The matrix $\Omega^{\star}$ is thus the product of three nonnegative matrices and thus it is also nonnegative. $\square$

**For Lemma 4.3**

*Proof of Lemma 4.3.* The proof of this lemma is similar to that of Lemma 3.3 with the optimal set is now given by $\mathcal{O}^{\star}(\beta) = \big\{\Omega \in \mathcal{U}_{\mathbb{W},\rho}(\widehat{\Omega}) : \langle \Omega, V(\beta)\rangle = F_{\mathbb{W},\rho}(\beta)\big\}$. The singleton of this $\mathcal{O}^{\star}(\beta)$ is guaranteed by Theorem 4.2. $\square$

## A.3   Discussions on Assumption 2.1.

Assumption 2.1(i) is standard in the machine learning literature and it holds naturally in many classification and regression tasks. Regarding Assumption 2.1(ii), the non-negativity of $\widehat{\Omega}$ follows directly if the weighting function $\omega$ is also non-negative. As shown below, positive definiteness holds under mild conditions of the training data and the weighting kernel.

**Lemma A.3** (Positive definite nominal weighting matrix). *Suppose that the weighting function $\omega$ can be represented by a strictly positive definite kernel $K : \mathcal{Z} \times \mathcal{Z} \to \mathbb{R}$ in the sense that $\omega(\widehat{z}_i) = K(z_0, \widehat{z}_i)$ for $i = 1, \ldots, N$ and for some distinct covariates $(z_0, \widehat{z}_1, \ldots, \widehat{z}_N) \in \mathcal{Z}^{N+1}$. Then*

$$\widehat{\Omega} \triangleq \begin{bmatrix} K(z_0, z_0) & K(z_0, \widehat{z}_1) & \cdots & K(z_0, \widehat{z}_N) \\ K(\widehat{z}_1, z_0) & K(\widehat{z}_1, \widehat{z}_1) & \cdots & K(\widehat{z}_1, \widehat{z}_N) \\ \vdots & \vdots & \ddots & \vdots \\ K(\widehat{z}_N, z_0) & K(\widehat{z}_N, \widehat{z}_1) & \cdots & K(\widehat{z}_N, \widehat{z}_N) \end{bmatrix}$$

*is positive definite with $\widehat{\Omega}_{0i} = \widehat{\Omega}_{i0} = \omega(\widehat{z}_i)$ for $i = 1, \ldots, N$.*

The proof of Lemma A.3 follows by noticing that $\widehat{\Omega}$ is the Gram matrix of a kernel $K$, and hence $\widehat{\Omega}$ is a positive definite matrix [4, pp. 2392].

The next result shows that given any weight $\omega(\widehat{z}_i)$, there exists a matrix $\widehat{\Omega}$ satisfying Assumption 2.1(ii). Notice that it is always possible to choose the numbers $d_k$ satisfying the specified conditions below.

**Lemma A.4** (Existence of $\widehat{\Omega}$). *Given any nonnegative weights $\omega(\widehat{z}_i)$ for $i = 1, \ldots, N$, let $A$ be a symmetric matrix with all entries zero except for those on the first row and first column, and on the diagonal as follows*

$$A \triangleq \begin{bmatrix} d_0 & \omega(\widehat{z}_1) & \cdots & \omega(\widehat{z}_N) \\ \omega(\widehat{z}_1) & d_1 & \cdots & 0 \\ \vdots & \vdots & \ddots & \vdots \\ \omega(\widehat{z}_N) & 0 & \cdots & d_N \end{bmatrix}.$$

*Then $A$ is positive definite and nonnegative if $d_k$ are chosen such that $d_0 > 0$, $d_1 \Delta_1 > \omega(\widehat{z}_1)^2$, and $d_k \Delta_k > d_1 \ldots d_{k-1} \omega(\widehat{z}_k)^2$ for $k = 2, \ldots, N+1$, where $\Delta_k$ denotes the $k$th leading principal minor of the matrix $A$ which is independent of $d_k, \ldots, d_{N+1}$.*

*Proof of Lemma A.4.* It is clear that $A$ is symmetric and nonnegative. The conditions on $d_k$ also ensures that $\Delta_k > 0$ for every $k = 1, \ldots, N+1$. Thus $A$ is positive definite as well. $\square$

For the given non-negative weights $\omega(\widehat{z}_i)$, Lemma A.4 shows that there are infinitely many doubly non-negative matrices $\widehat{\Omega}$ that satisfy the condition $\widehat{\Omega}_{0i} = \widehat{\Omega}_{i0} = \omega(\widehat{z}_i)$ for $i = 1, \ldots, N$.

In practice, the choice of $\widehat{\Omega}$ can impact the performance of our robust estimate, and fine-tuning the elements of $\widehat{\Omega}$ may improve the predictive power. For the scope of this paper, we aim to improve the robustness of NW and LLR estimators and therefore will mainly focus on the scenario described in Lemma A.3 where $\widehat{\Omega}$ is given by a strictly positive definite kernel.

# B   Implementation Details

## B.1   Gradient Information

We provide here the gradient information for the convex problems (8) and (12). This information can be exploited to derive fast numerical routines to find the optimal dual solution $\gamma^\star$.

Denote by $g$ the objective function of problem (8). The gradient of $g$ is

$$\nabla g(\gamma) = \rho - \log \det(I - \gamma^{-1} \widehat{\Omega}^{\frac{1}{2}} V(\beta) \widehat{\Omega}^{\frac{1}{2}}) - \frac{1}{\gamma} \big\langle (I - \gamma^{-1} \widehat{\Omega}^{\frac{1}{2}} V(\beta) \widehat{\Omega}^{\frac{1}{2}})^{-1}, \widehat{\Omega}^{\frac{1}{2}} V(\beta) \widehat{\Omega}^{\frac{1}{2}} \big\rangle.$$

Let $h$ be the objective function of problem (12). The gradient of $h$ is

$$\nabla h(\gamma) = \rho - \big\langle \widehat{\Omega}, (I - \gamma^\star [\gamma^\star I - V(\beta)]^{-1})^2 \big\rangle.$$

## B.2   Implementation

Following Lemma 3.5, for a fixed value of $\beta$, we can rewrite the *low-rank* matrix $V(\beta)$ using the eigenvalue decomposition $V(\beta) = Q\Lambda Q^\top$, where $\Lambda \in \mathbb{R}^{2 \times 2}$ is a diagonal matrix and $Q \in \mathbb{R}^{(N+1) \times 2}$ is an orthonormal matrix. Therefore, we can leverage the *Woodbury matrix identity* to implement the inverse in gradient computation (e.g., $\nabla g$ and $\nabla h$) efficiently. For examples, (i)

$$(\gamma I - V(\beta))^{-1} = \gamma^{-1} I - \gamma^{-2} Q \left( -\Lambda^{-1} + \gamma^{-1} I_2 \right)^{-1} Q^\top,$$

where we have exploited the fact that $Q^\top Q = I_2$ with $I_2$ being the $2 \times 2$ identity matrix; and (ii)

$$(I - \gamma^{-1} \widehat{\Omega}^{\frac{1}{2}} V(\beta) \widehat{\Omega}^{\frac{1}{2}})^{-1} = I - \widehat{\Omega}^{\frac{1}{2}} Q \left( -\gamma \Lambda^{-1} + Q^\top \widehat{\Omega} Q \right)^{-1} Q^\top \widehat{\Omega}^{\frac{1}{2}}.$$

Therefore, the inverse in gradient (e.g., $\nabla g$ and $\nabla h$) is computed on $2 \times 2$ matrices.

# C   Additional Empirical Results

**Details of datasets.**   Table A.1 lists the detailed statistical characteristics of the datasets used in our experiments.

Table A.1: Statistical characteristics of the datasets.

|          | Abalone | Bank | CPU  | Elevators | KIN   | POL   | Puma | Slice |
|----------|---------|------|------|-----------|-------|-------|------|-------|
| #samples | 4177    | 8192 | 8192 | 16599     | 40000 | 15000 | 8192 | 53500 |
| #features | 7      | 32   | 21   | 17        | 8     | 26    | 32   | 384   |

**Further results for the ideal case: no sample perturbation.** We illustrate further empirical results for *all* 8 *datasets* and with different nearest neighbor size $N$ (e.g., similar to Figure 1 with the nearest neighbor size $N = 50$).

- For $N = 50$, we illustrate results in Figure A.1.

- For $N = 30$, we illustrate results in Figure A.2.

- For $N = 20$, we illustrate results in Figure A.3.

- For $N = 10$, we illustrate results in Figure A.4.

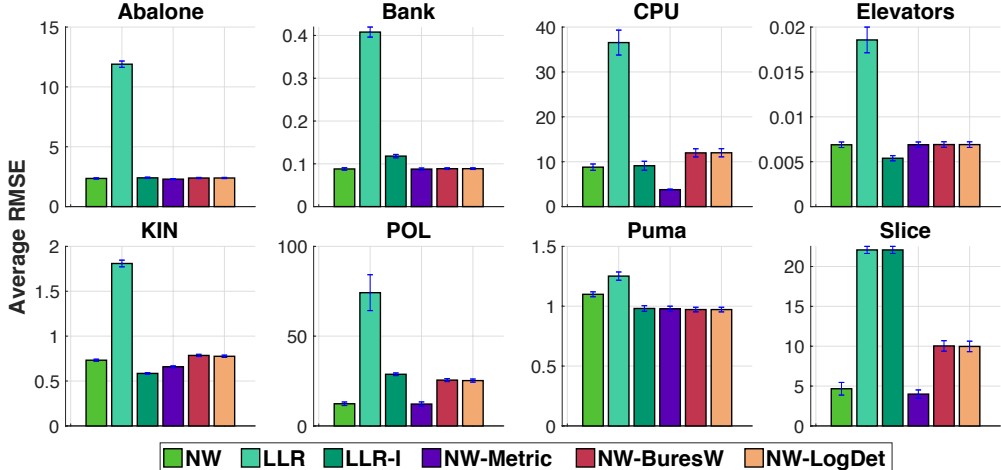

Figure A.1: Average RMSE for ideal case with no perturbation when $N = 50$ for all 8 datasets.

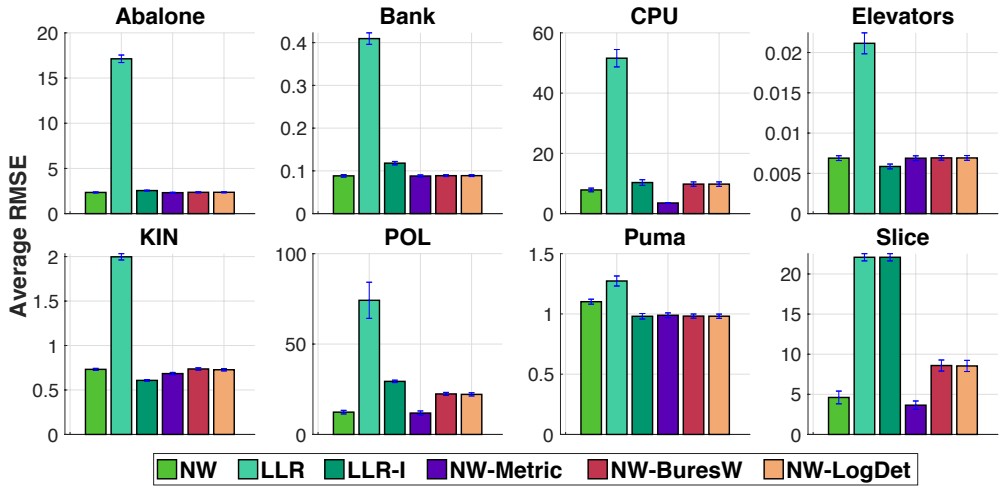

Figure A.2: Average RMSE for ideal case with no perturbation when $N = 30$ for all 8 datasets.

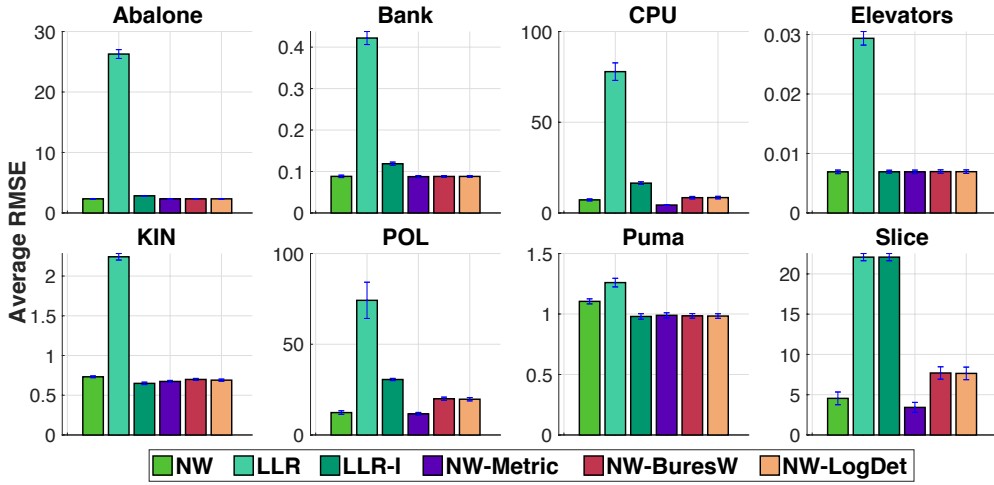

Figure A.3: Average RMSE for ideal case with no perturbation when $N = 20$ for all $8$ datasets.

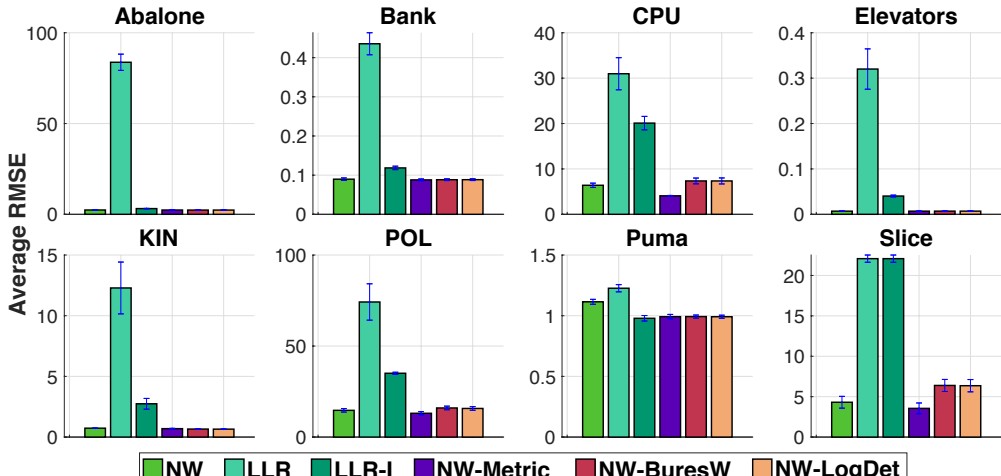

Figure A.4: Average RMSE for ideal case with no perturbation when $N = 10$ for all $8$ datasets.

**Further results for the cases when training samples are perturbed.** We illustrate further empirical results for *all 8 datasets* with different nearest neighbor size $N$ (e.g., similar to Figure 2 where the nearest neighborr size $N = 50$).

- For $N = 50$, we illustrate results in Figure A.5.

- For $N = 30$, we illustrate results in Figure A.6.

- For $N = 20$, we illustrate results in Figure A.7.

- For $N = 10$, we illustrate results in Figure A.8.

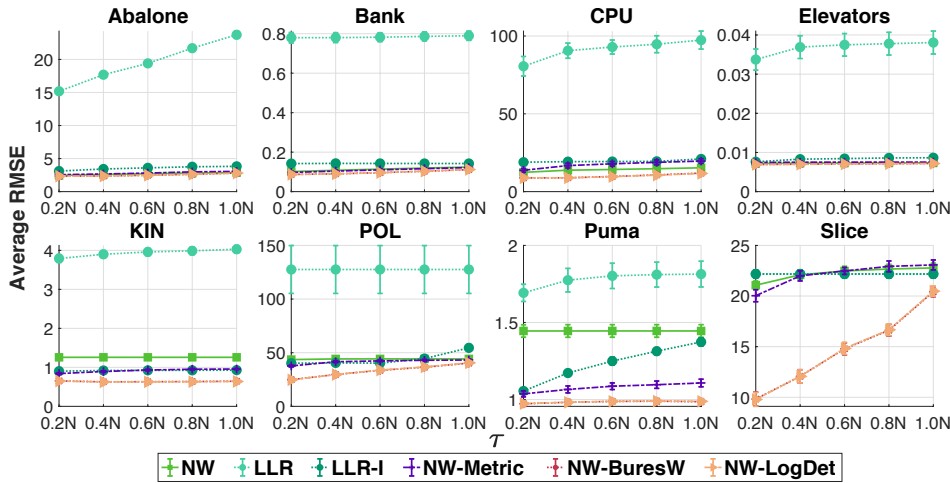

Figure A.5: RMSE for varying perturbation levels $\tau$ where $N = 50$.

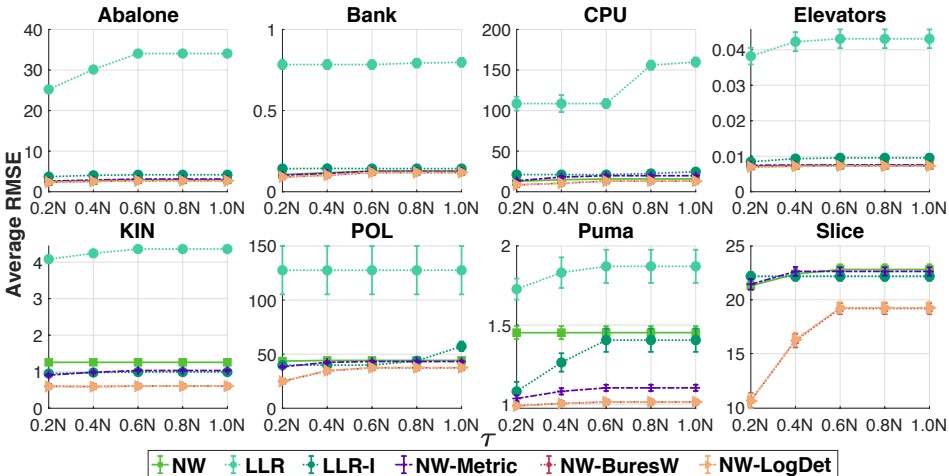

Figure A.6: RMSE for varying perturbation levels $\tau$ where $N = 30$.

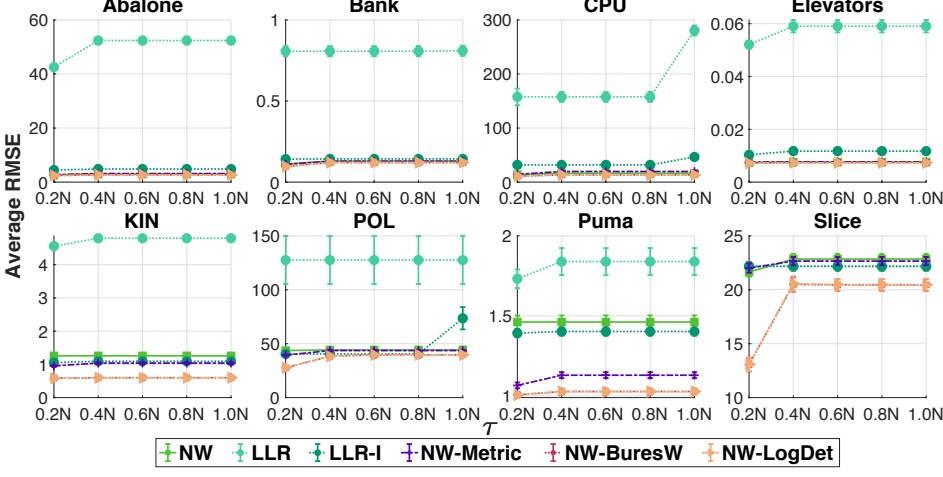

Figure A.7: RMSE for varying perturbation levels $\tau$ where $N = 20$.

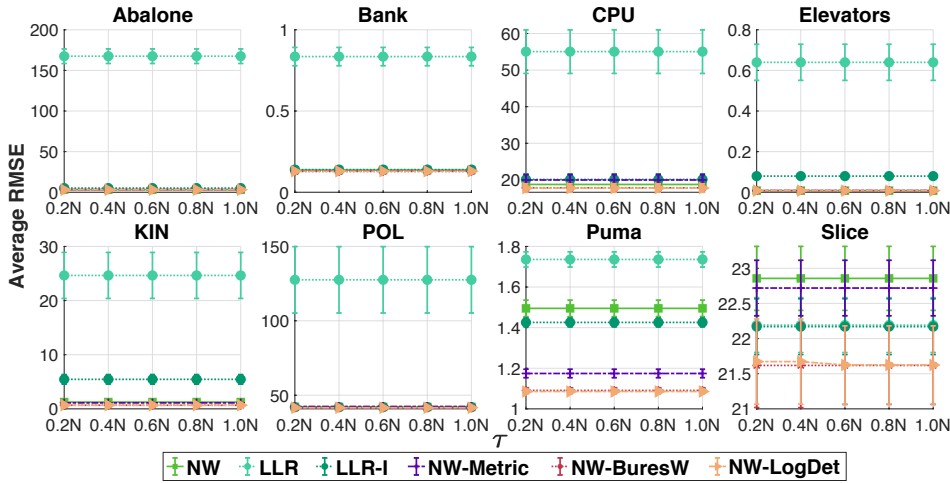

Figure A.8: RMSE for varying perturbation levels $\tau$ where $N = 10$.

**Further results for the effects of the uncertainty size $\rho$.** We illustrate further empirical results for different nearest neighbor size $N$ in *all 8 datasets* (e.g., similar to Figure 3 where the nearest neighbor size $N = 50$ and the perturbation $\tau = N$). Note that when $\rho = 0$, the reweighting schemes are equivalent to the vanilla NW estimator.

- For $N = 50$, we illustrate results for NW-LogDet and NW-BuresW in Figure A.9 and Figure A.10 respectively.

- For $N = 30$, we illustrate results for NW-LogDet and NW-BuresW in Figure A.11 and Figure A.12 respectively.

- For $N = 20$, we illustrate results for NW-LogDet and NW-BuresW in Figure A.13 and Figure A.14 respectively.

- For $N = 10$, we illustrate results for NW-LogDet and NW-BuresW in Figure A.15 and Figure A.16 respectively.

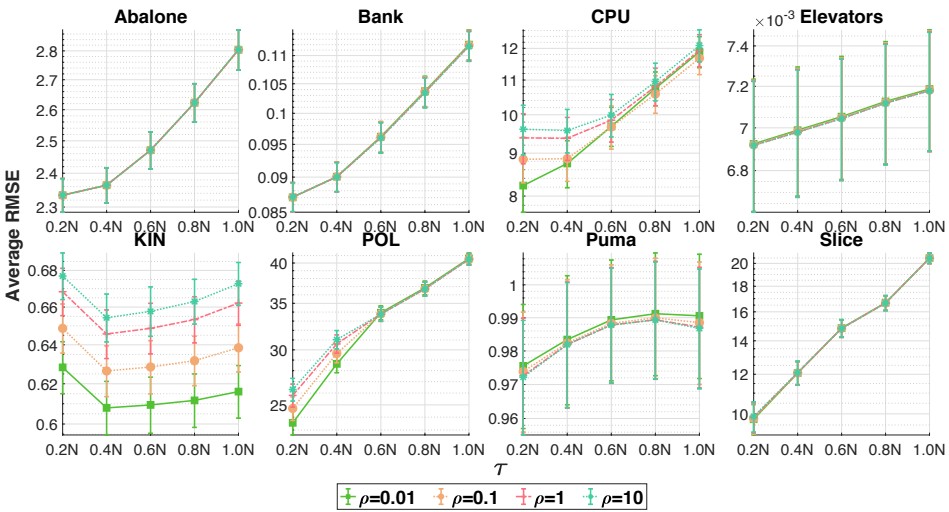

Figure A.9: Effects of the uncertainty size $\rho$ on RMSE for NW-LogDet when $N = 50$.

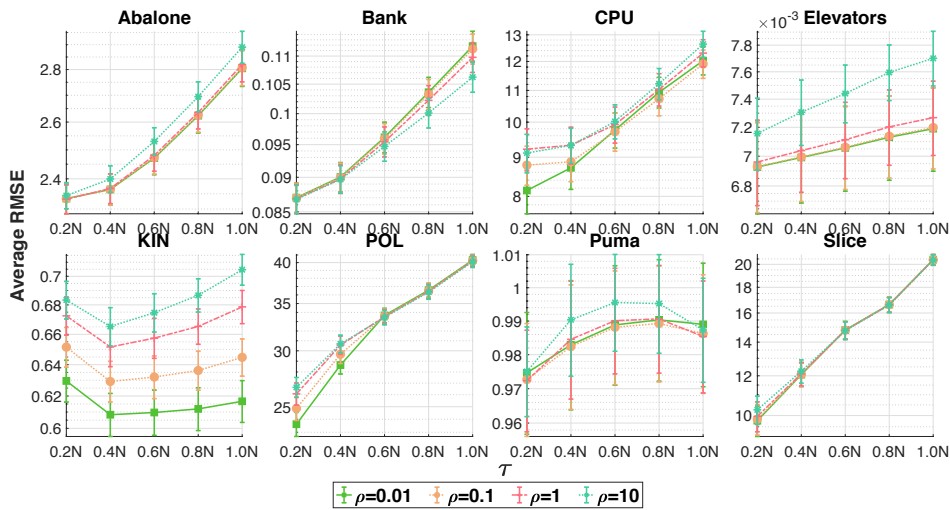

Figure A.10: Effects of the uncertainty size $\rho$ on RMSE for NW-BuresW when $N = 50$.

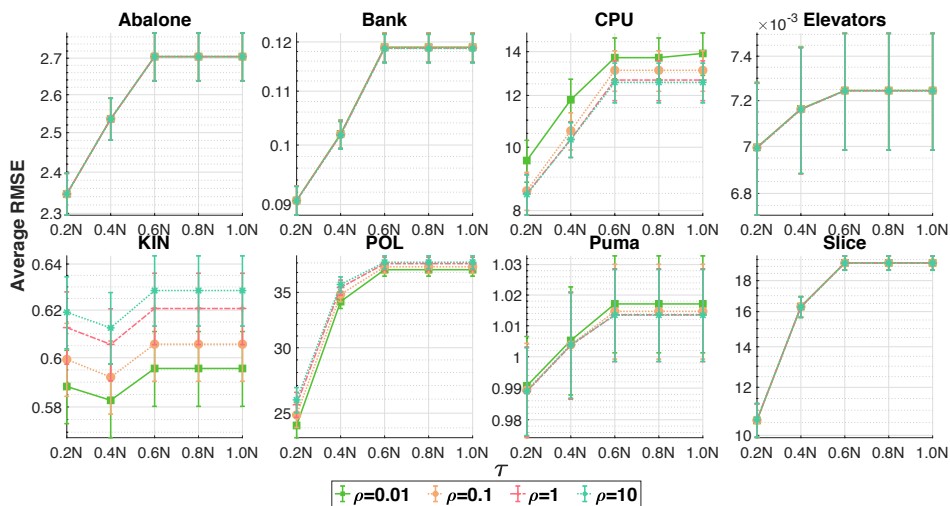

Figure A.11: Effects of the uncertainty size $\rho$ on RMSE for NW-LogDet when $N = 30$.

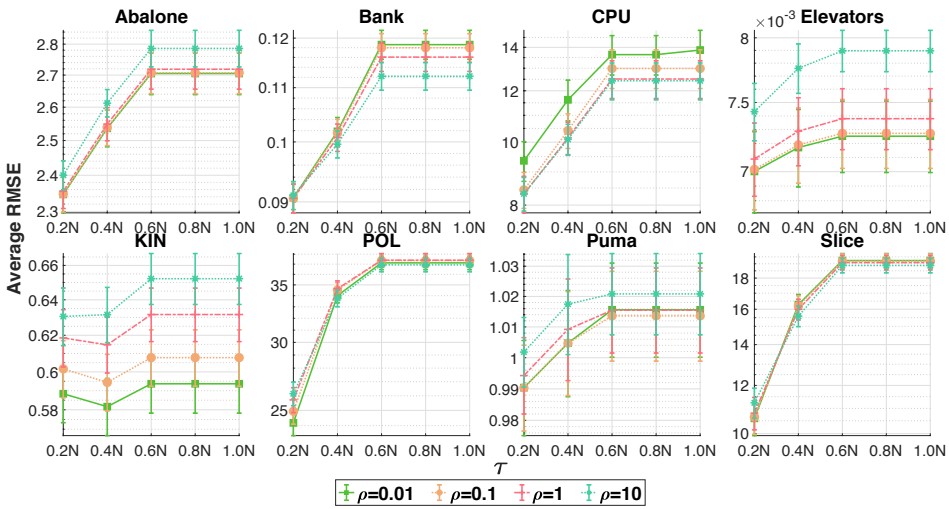

Figure A.12: Effects of the uncertainty size $\rho$ on RMSE for NW-BuresW when $N = 30$.

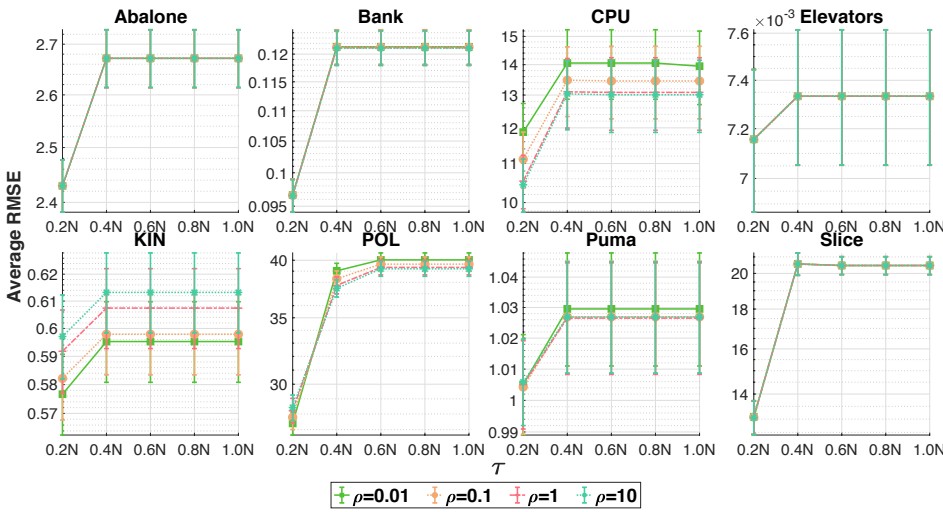

Figure A.13: Effects of the uncertainty size $\rho$ on RMSE for NW-LogDet when $N = 20$.

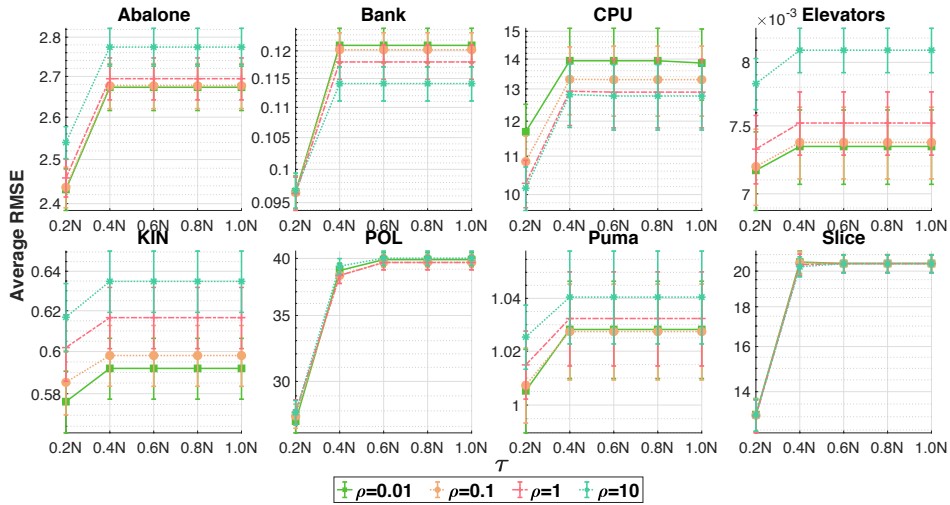

Figure A.14: Effects of the uncertainty size $\rho$ on RMSE for NW-BuresW when $N = 20$.

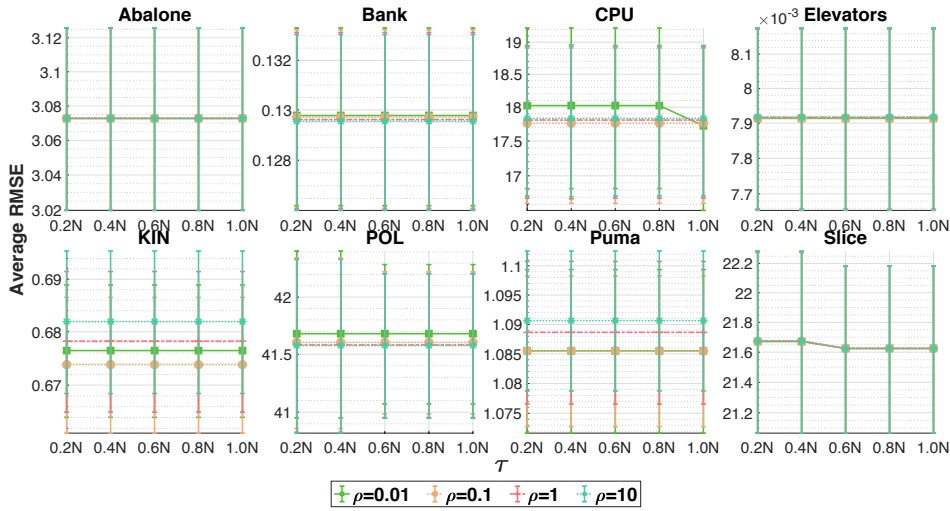

Figure A.15: Effects of the uncertainty size $\rho$ on RMSE for NW-LogDet when $N = 10$.

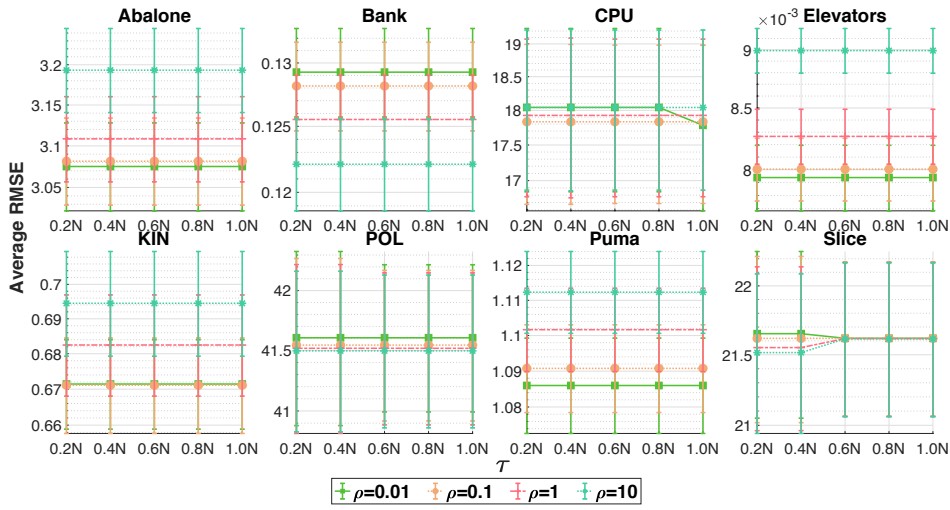

Figure A.16: Effects of the uncertainty size $\rho$ on RMSE for NW-BuresW when $N = 10$.

**Further results for the effects of the nearest neighbor size $N$.** We illustrate further empirical results for different uncertainty size $\rho$ in *all 8 datasets* (e.g., similar to Figure 4 where $\rho = 0.1$ in the KIN dataset).

- For $\rho = 10$, we illustrate results for NW-LogDet and NW-BuresW in Figure A.17 and Figure A.18 respectively.

- For $\rho = 1$, we illustrate results for NW-LogDet and NW-BuresW in Figure A.19 and Figure A.20 respectively.

- For $\rho = 0.1$, we illustrate results for NW-LogDet and NW-BuresW in Figure A.21 and Figure A.22 respectively.

- For $\rho = 0.01$, we illustrate results for NW-LogDet and NW-BuresW in Figure A.23 and Figure A.24 respectively.

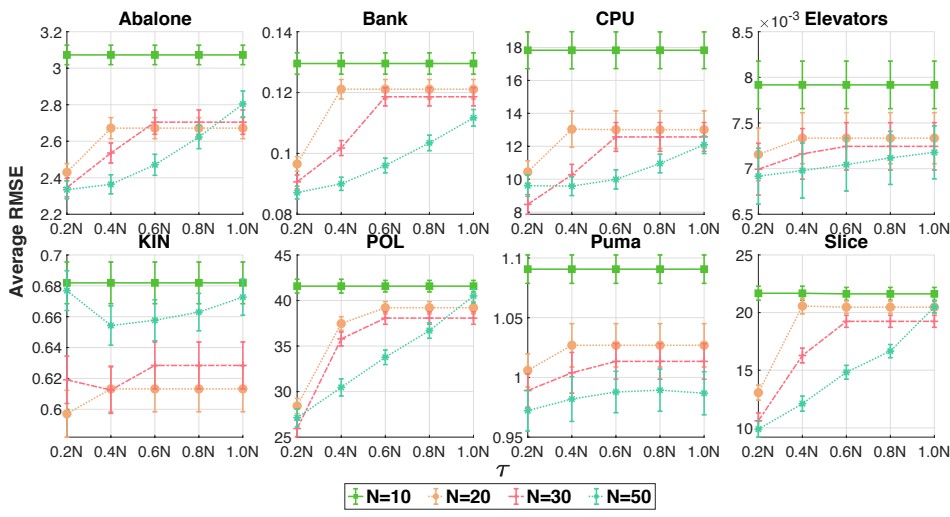

Figure A.17: Effects of the nearest neighbor size $N$ on RMSE for NW-LogDet when $\rho = 10$.

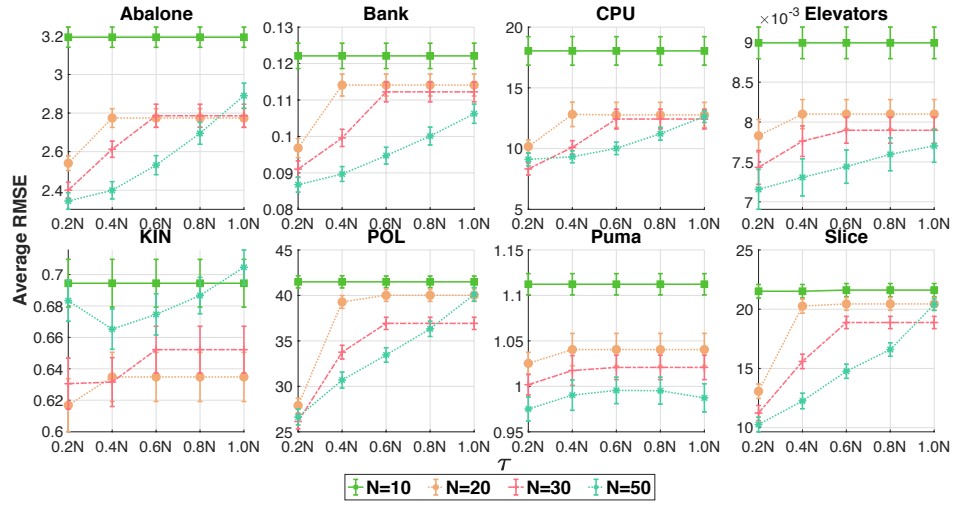

Figure A.18: Effects of the nearest neighbor size $N$ on RMSE for NW-BuresW when $\rho = 10$.

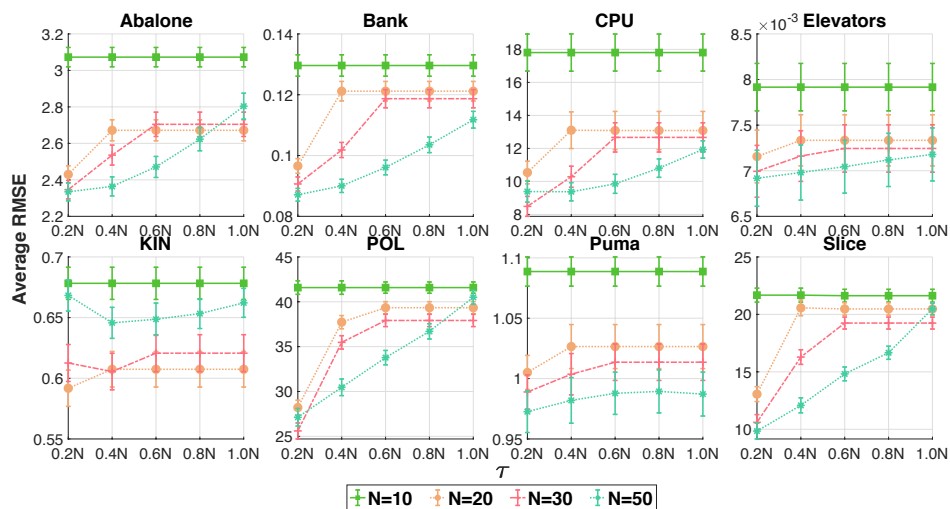

Figure A.19: Effects of the nearest neighbor size $N$ on RMSE for NW-LogDet when $\rho = 1$.

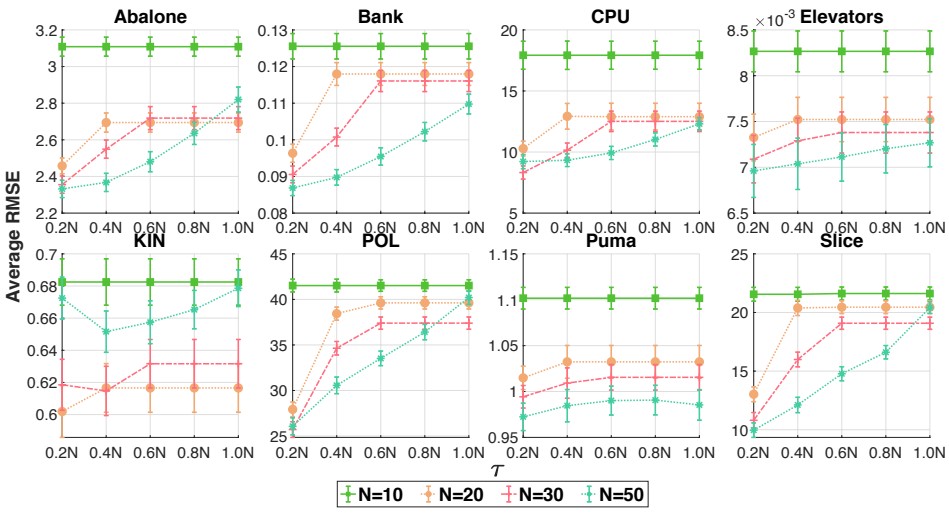

Figure A.20: Effects of the nearest neighbor size $N$ on RMSE for NW-BuresW when $\rho = 1$.

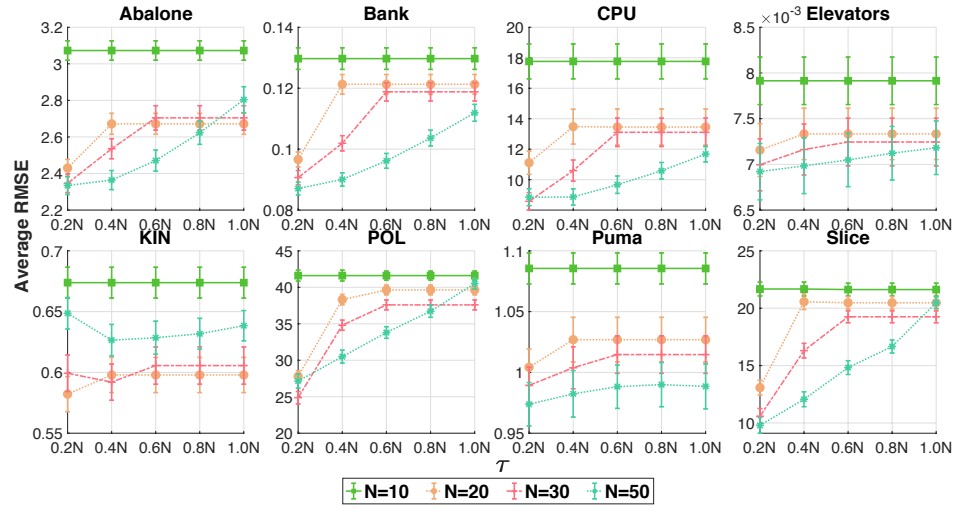

Figure A.21: Effects of the nearest neighbor size $N$ on RMSE for NW-LogDet when $\rho = 0.1$.

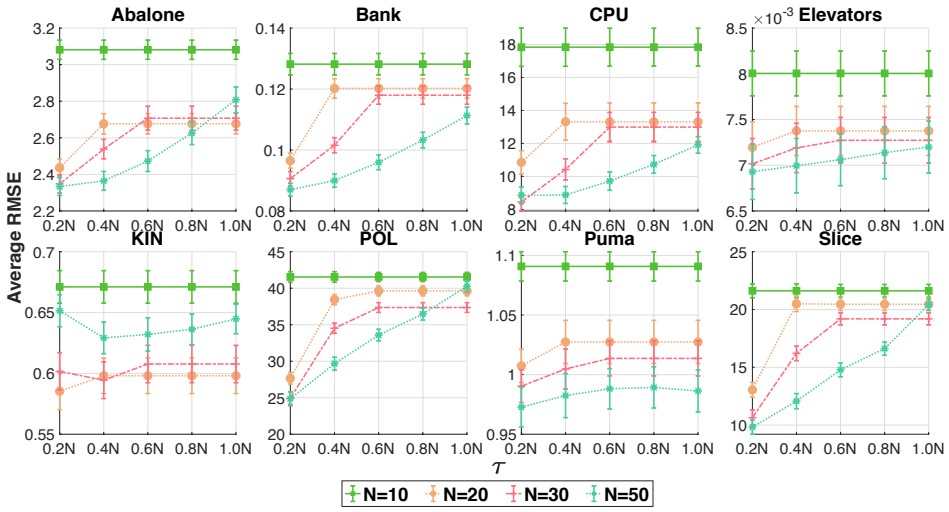

Figure A.22: Effects of the nearest neighbor size $N$ on RMSE for NW-BuresW when $\rho = 0.1$.

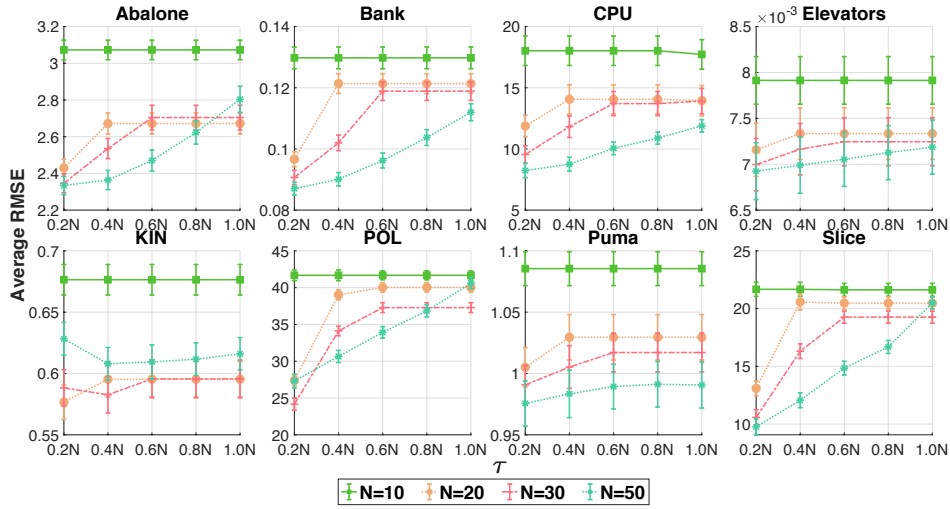

Figure A.23: Effects of the nearest neighbor size $N$ on RMSE for NW-LogDet when $\rho = 0.01$.

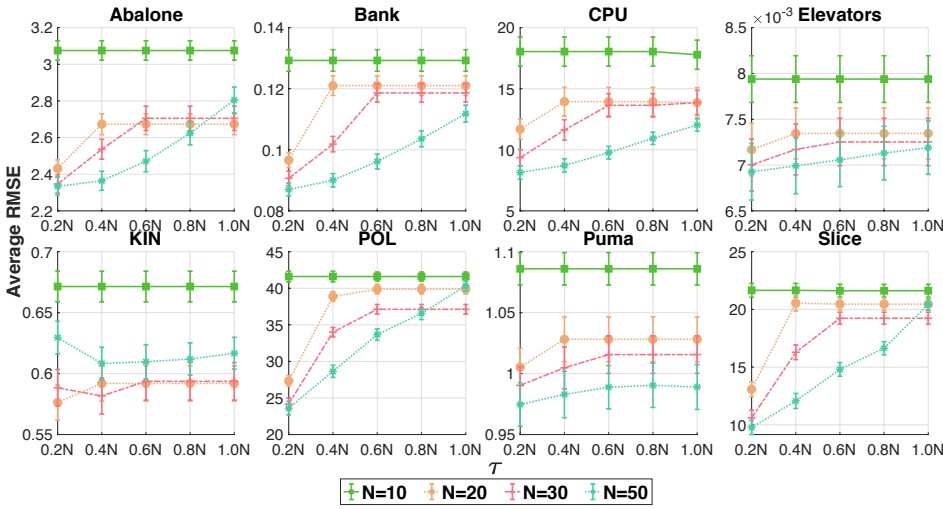

Figure A.24: Effects of the nearest neighbor size $N$ on RMSE for NW-BuresW when $\rho = 0.01$.

**Further results for varying shifting w.r.t.** $\kappa$. We first illustrate corresponding results for NW-BuresW in Figure A.25, similar as results in Figure 5 for NW-LogDet.

We next illustrate further empirical results for different uncertainty size $\rho$ and different perturbation $\tau$ in the KIN dataset (e.g., similar to Figure 5 and Figure A.25 where $\rho = 0.1$ for the left plots and $\tau = 0.2N$ for the right plots).

- For NW-LogDet, we illustrate results on effects of perturbation intensity $\kappa$ for different perturbation proportion $\tau$ in Figure A.26 and Figure A.27 respectively.

- For NW-BuresW, we illustrate results on effects of perturbation intensity $\kappa$ for different perturbation proportion $\tau$ in Figure A.28 and Figure A.29 respectively.

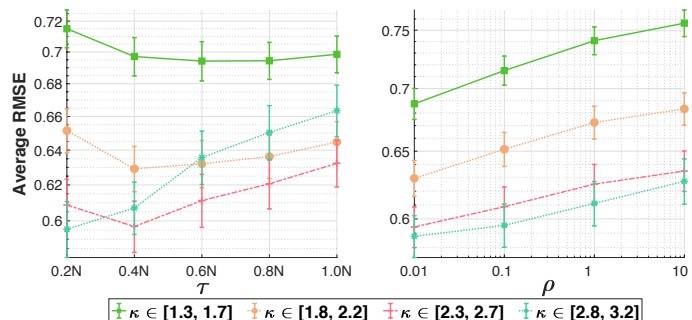

Figure A.25: Effects of perturbation intensity $\kappa$ for NW-BuresW estimate. Left plot: different perturbation $\tau$, right plot: different uncertainty size $\rho$.

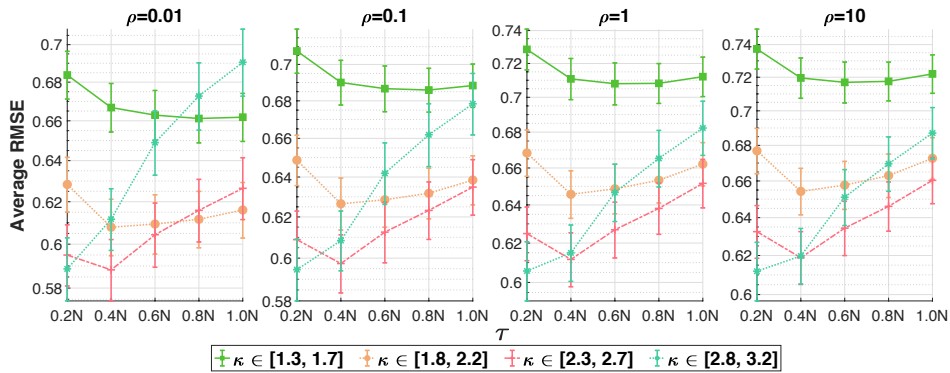

Figure A.26: Effects of perturbation intensity $\kappa$ for NW-LogDet estimate for different perturbation proportion $\tau$.

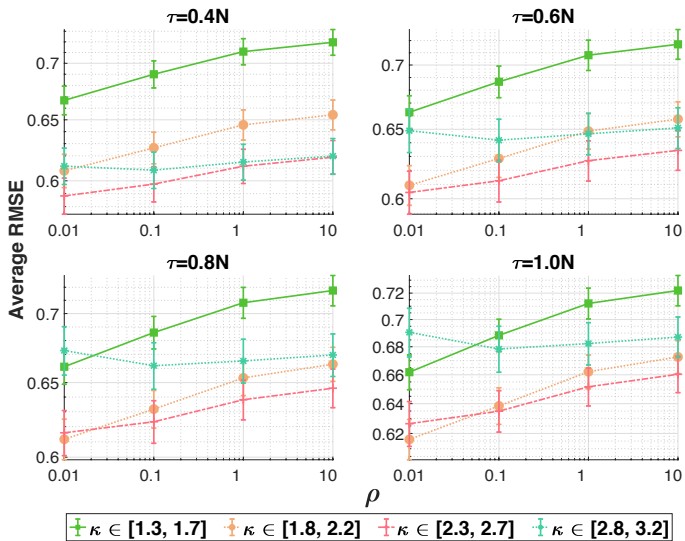

Figure A.27: Effects of perturbation intensity $\kappa$ for NW-LogDet estimate for different uncertainty size $\rho$.

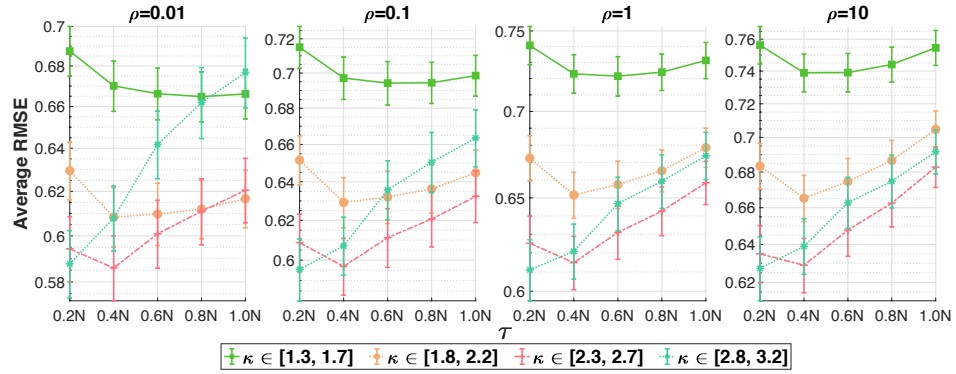

Figure A.28: Effects of perturbation intensity $\kappa$ for NW-BuresW estimate for different perturbation proportion $\tau$.

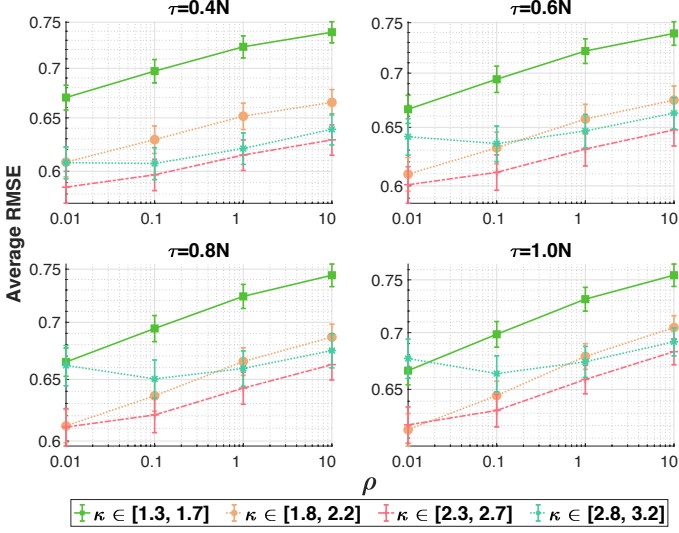

Figure A.29: Effects of perturbation intensity $\kappa$ for NW-BuresW estimate for different uncertainty size $\rho$.

**Further results for effects of $N$ on computational time.** We illustrate further effects of $N$ on computational time in *all 8 datasets* (e.g., similar to Figure 6 for the KIN dataset).

- For NW-LogDet, we illustrate further effects of $N$ on computational time in all 8 datasets in Figure A.30.
- For NW-BuresW, , we illustrate further effects of $N$ on computational time in all 8 datasets in Figure A.31.

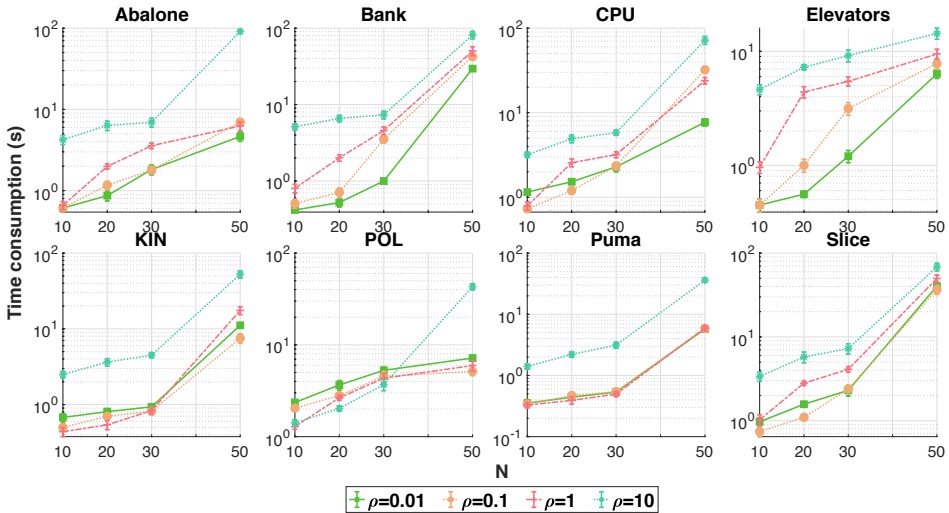

Figure A.30: Effects of $N$ on computational time for NW-LogDet.

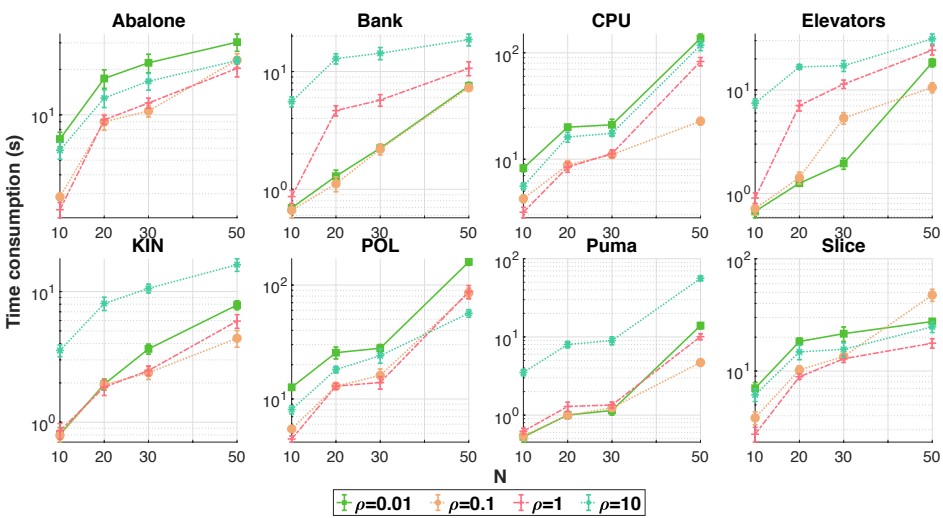

Figure A.31: Effects of $N$ on computational time for NW-BuresW.