# OpenReview forum: "Adversarial Regression with Doubly Non-negative Weighting Matrices"
_NeurIPS.cc/2021/Conference — NeurIPS 2021 Poster_

### Official Review · Reviewer_rPg1 · 2021-07-16

**Rating:** 7
**Confidence:** 4

**Summary:**

The paper attempts an important problem of learning a regression model in the presence of adversarial perturbation. The paper formulates a general class of problems and proposes a solution for that. Finally, experimental results are also provided.



**Ethical Concerns:**

I think there is no risk of ethical concerns in this paper.


**Limitations And Societal Impact:**

The authors did not discuss this aspect of the paper. However, I think they can connect their work to privacy as I suggested above and in turn can be connected to the privacy aspect of society and AI.

**Main Review:**

Overall, the paper seems to be technically good. The necessary technical analysis is provided. However, it could be simplified further to make it readable to a wider range of readers. For example, Eq. 1 and associated text lack enough clarity without a context which is more widely known. Introduction of z and parametrizing the weights with that can be a bottleneck for many readers. Description in simple words or simpler examples will improve the paper.  Similar comments hold for motivation of the problem and intuition of the method. A precise problem definition comes much later than too much technical nitty gritty. Otherwise, the technical quality is sound and significant.

Formulation of reweighting in terms of non-negative matrices is interesting and could make it possible to derive analytical results which were otherwise difficult. This part of the paper is original. However, connecting to standard perturbation techniques such as differential privacy could make the paper more interesting to read and connect to the privacy community as well.

Quality of the paper is limited to the technical analysis but can be adequate. The biggest drawback of the paper is lack of clarity which is discussed above.
========================
I thank author for their answer to my comment. I shall go with my previous score hope author will simplify the introduction as promised.

**Time Spent Reviewing:**

6

---

> ### Author Response · Authors · 2021-08-10
> **Response to Reviewer rPg1**
>
> We thank the reviewer for the appreciation of our paper. We will try to improve the exposition of the paper, especially in the motivation of the problem.
>
> The goal of equation (1) was to introduce the most general form of the problem. But we agree that it can be a big jump right at the beginning of the paper. We will rewrite the introduction accordingly.
>
> We also thank the reviewer for the connection with privacy and the societal impact of AI. We believe that it is an important problem and there are numerous connections to be made to improve that current landscape of AI. Thank you!

---

### Official Review · Reviewer_sDqY · 2021-07-16

**Rating:** 6
**Confidence:** 3

**Summary:**

This proposes a new method for kernel-reweighted regression by reparametrizing the sample weights using a doubly non-negative matrix. When the weighting matrix is confined in an uncertainty set using either the log-determinant divergence or the Bures Wasserstein distance (defined on psd matrices), it is shown that that the adversarially reweighted estimate can be solved efficiently using first-order methods. Some experiments are presented.


**Limitations And Societal Impact:**

Weak points
---
- Work seems very incremental, and the contributions don't appear to be sufficient for publication at a conference like NeurIPS.

- Experiments are not inclusive. All what the experiments seem to show is that LLR (local linear regression) is much worst than the other estimators (see Figure 1). For example, the experiments don't even show that NW-LogDet / NW-BuresW are any better than plane NW. So the question then is what is the advantage, theoretical or algorithmic, of the proposed estimators.

- Missing statistical analysis of the resulting estimator.

- There is a large body of work on distributional robust optimization out there. This work fits into that space, but unfortunately, the authors make no links.

**Main Review:**

Strengths
---
- Paper is well-written and easy to follow.
- The reduction of the adversarial problem to a much simpler one-dimensional problem is appealing


**Time Spent Reviewing:**

2

---

> ### Author Response · Authors · 2021-08-10
> **Response to Reviewer sDqY**
>
> We thank the reviewer for your feedback.
>
> **Q1**: We strongly disagree with your judgement that the contribution of this paper is incremental. We would like to encourage the reviewer to review the introduction of this paper where we have:
>
> - (i) clearly stated the difficulty of the problem,
>
> - (ii) indicated that there is currently no approach that can perform robustification (using the min-max formulation),
> In this paper, we have provided all the necessary ingredients to solve this open problem with both mathematical analysis and algorithmic routines.
>
> **Q2**: We respectfully disagree with your judgement. As in line 272-275, when we have the ideal case: no sample perturbation, the proposed method performs competitively against other baselines. However, as in line 281-294, when training samples are perturbed, the performance of the NW, LLR, LLR-I, NW-Metric baselines severely deteriorate while both NW-LogDet and NW-BuresW can alleviate the effect of data perturbation (as we discuss in line 56-64).
>
> We will follow the suggestion of **Reviewer rsLw** to remove the LLR and replot Figure 2 so that the performance gap between other 4 methods becomes more clear (and move the current Figure 2 with all baselines for completeness into the supplementary). We will clarify this point.
>
> **Q3**: Our paper focuses on the computational aspect of the estimator with robust reweighting, and we provide in the paper the detailed analysis of how this estimator is computed. We agree that the statistical property of this estimator is an interesting problem to study, however, we feel that it lies beyond the scope of this paper.
>
> **Q4**: We agree with the reviewer that the paper is closely related to the distributionally robust optimization field. This relation is clearly reflected in the literature review where we have mentioned relevant papers: References [9, 14, 18, 19, 36] are all in the field of distributionally robust optimization. Other references such as [12, 30] are also related to distributionally robust optimization to a certain extent.

---

> > ### Comment · Reviewer_sDqY · 2021-09-01
> > **Update**
> >
> > Thanks for the response. I still think the paper is incremental. Also, lack of statistical understanding of resulting estimator is not very satisfying. I'm convinced by the response given by the authors to my other points. Thus, I have updated my score from 5 to 6.

---

### Official Review · Reviewer_rsLw · 2021-07-17

**Rating:** 7
**Confidence:** 2

**Summary:**

$\textbf{Summary and main contributions:}$

This paper aims to improve the performance of the weighted regression model in the scenario where data samples are under covariate perturbations. The main contributions are summarized below:

(1) The authors propose to use doubly nonnegative matrices to reparameterize the sample weights in the kernel-reweighted regression model.

(2) For the set of feasible weighting matrices, in the view of Log-Determinant divergence and Bures-Wasserstein type divergence, the authors theoretically show that the proposed kernel-reweighted regression model with doubly nonnegative weighting matrices is efficiently solvable.

(3) Extensive experiment results on several datasets demonstrate its effectiveness under certain levels of sample perturbation.

**Limitations And Societal Impact:**

The limitations of the proposed method are not adequately discussed in the paper. Besides, it would be better if the authors explain more detailed differences between the proposed method and [1].

$\textbf{References:}$

[1] Yung-Kyun Noh, Masashi Sugiyama, Kee-Eung Kim, Frank Park, and Daniel D Lee. Generative local metric learning for kernel regression. In Advances in Neural Information Processing 417 Systems, pages 2452–2462, 2017.

--------------------------------$\textbf{Post-rebuttal}$--------------------------------

Thanks the authors for the detailed explanations.  I raise my score from 6 to 7 for the following two reasons:

1. Most of my concerns are well addressed. Although for my second additional comment, I still hold the view that the proposed method might be less effective in the setting when there is no perturbation or slight perturbation. This may be viewed as a potential (and acceptable) weakness since in real-world scenarios, we may not have access to the perturbation level.

2. I didn't observe any crucial weaknesses from other reviewers' feedback that convince me to change my attitude from acceptance to rejection.

Best

Reviewer rsLw

**Main Review:**

$\textbf{Clarity and originality:}$ the paper is well written. The proposed kernel-reweighted regression model with doubly nonnegative weighting matrices is a novel contribution.

$\textbf{Significance:}$ when trained with perturbed samples, NW-LogDet is clearly effective and outperforms other methods under varying perturbation levels and uncertainty size. It has also been shown that the adversarially reweighted estimate is solvable with first-order methods.

$\textbf{Additional comments and suggestions:}$

(1) The motivation to the use of Log-Determinant divergence and Bures-Wasserstein type divergence is not intuitive. It would be better if the authors add additional explanations regarding the insights of these two divergences.

(2) One weakness of the proposed method is that it is less effective in the setting when there is no perturbation. As depicted in Figure 1, NW-metric appears like a more competitive method. In practice, it is likely that we do not have the knowledge of perturbation level, as a result, always adopting the proposed method may not be a good choice.

(3) In figure 2, it would be better if the authors remove  LLR so that the performance gap between all other four methods would be more clear.

**Time Spent Reviewing:**

>6 hours

---

> ### Author Response · Authors · 2021-08-10
> **Response to Reviewer rsLw**
>
> We thank the reviewer for your positive comments.
>
> > (1) The motivation to the use of Log-Determinant divergence and Bures-Wasserstein type divergence is not intuitive. It would be better if the authors add additional explanations regarding the insights of these two divergences.
>
> Both the Log-Determinant divergence and Bures-Wasserstein type divergence are popular geometry and wide-used in practice for positive definite matrices. Moreover, by leveraging those divergences, it also allows us to derive efficient algorithmic approaches based on the first-order methods for our proposed adversarial reweighting schemes in the Problem (4).
>
> Notice that one is tempted to try other metrics on the space of matrices: for example, one can build the uncertainty set using the Frobenius norm or any other matrix norm. Nevertheless, these choices will **not** lead to an analytical formula for $\Omega^*$ as is reported in Theorem 3.2 and 4.2. The advantage of using the Log-Determinant and the Bures-Wasserstein divergence is that they ensure the weighting matrix to be positive definite in a natural manner. (Note that Frobenius norm, for example, does not require the matrices to be positive definiteness.)
>
> > (2) One weakness of the proposed method is that it is less effective in the setting when there is no perturbation. As depicted in Figure 1, NW-metric appears like a more competitive method. In practice, it is likely that we do not have the knowledge of perturbation level, as a result, always adopting the proposed method may not be a good choice.
>
> As in line 272-275, for the ideal case with no sample perturbation, our adversarial reweighting schemes perform competitively against the baselines on several datasets (as in Figure 1 for 4 datasets, and Figure A.1 in the supplementary for all 8 datasets). We agree that NW-metric appears like a more competitive method. However, as in line 281-294, when training data are perturbed, the performances of the NW, LLR, LLR-I and NW-Metric baselines severely deteriorate while the proposed adversarial reweighting schemes (NW-LogDet and NW-BuresW) can alleviate the effect of data perturbation. (See Figure 2 for 4 datasets, and Figure A.5 in the supplementary for all 8 datasets).
>
> We would like to emphasize that the data can be easily perturbed in reality. For example, to promote privacy when the response variable carries sensitive information, it is advisable to add a multiplicative noise to the data. This corresponds to the setup that we test in the numerical experiments.
>
> Notice that when there is no perturbation, the drop in performance for the robust approach is small (e.g., the robust approaches are comparative to NW-Metric on 4/8 datasets). However, when there is a perturbation, the robust approach clearly outperforms the NW-estimator (e.g., the robust approaches outperform other baselines on *all 8 datasets*, especially on Slice and Puma datasets). This suggests that the robust approach should be encouraged.
>
> > (3) In figure 2, it would be better if the authors remove LLR so that the performance gap between all other four methods would be more clear.
>
> The current Figure 2 is reported for the purpose of completeness with all the baselines. We will follow your suggestion to replot Figure 2 and move the current Figure 2 into the supplementary.
>
> > it would be better if the authors explain more detailed differences between the proposed method and [1] (Noh et al.).
>
> Recall that the NW-Metric [Noh et al.] is a metric learning approach that learns the Mahalanobis distance from training data points to replace the Euclidean distance in the Gaussian kernel used in the NW estimator. Our method, however, does not learn a metric. We focus on the weight values, which enter the problem more explicitly via equation (1). We will clarify this point.

---

### Official Review · Reviewer_cyV5 · 2021-07-17

**Rating:** 8
**Confidence:** 3

**Summary:**

This paper proposes a novel methodology for constructing robust estimates for kernel regression. This method works for methods that can be cast as locally-reweighted regression problems, such as the Nadaraya-Watson estimator and local polynomial regression in general. The methodology consists of viewing the traditional optimization objective as a vector inner product and then lifting to a matrix inner product. One can then define uncertainty sets (in the optimization sense) over which we would like to minimize the maximum of a matrix inner product using matrix divergences such as the log-determinant divergence and the Bures-Wasserstein distance. The paper then shows that it is possible to efficiently solve these problems.

**Limitations And Societal Impact:**

On the other hand, I feel like there are some big unanswered questions. Why should we care about this particular formulation. Does this correspond to anything meaningful for a nonparametric regression problem? I know there are some short discussions on the particular divergences as related to normal random variables, but I don’t really understand what this is doing for my regression problem and why I should use it.

Other issues that are more technical in nature get magnified by the fact that I don’t know what this robustification means for my regression problem. In Section 2, line 105, the discussion on the values of $\hat{\Omega}$ is somewhat limited. On a first read of the paper, I thought the unspecified entries were 0, which would make the matrix positive semi-definite and therefore incompatible with the log-determinant divergence robustness. On taking another look, it became clear that one can pick the other entries of the matrix so as to make it positive semi-definite. But, almost certainly this choice is not unique. So, again, what does it do to my nonparametric regression problem to choose different matrices $\hat{\Omega}$?

A much more complete analysis of this problem would relate the proposed approaches to the regression problem. The rest of the paper is strong enough to be a clear accept if it had this as well.



**Main Review:**

Overall, I like this paper in a lot of ways. The new optimization problem is interesting in its own right. The fact that there are some matrix divergences for which the problem may be solved is also interesting, and it looks like some non-trivial effort goes into these proofs. The paper is also fairly clear and easy to read.


------Post-Rebuttal------

I've changed my mind about this paper. I've mostly changed my mind because (a) I don't believe any other reviewers have brought up any major concerns and (b) I feel that this is a fairly creative paper.

I am disappointed that the authors seem unable to justify the proposed approach formally, but perhaps this was partly my error in communicating my concern. Let me try again. I tried to draw a diagram, and it didn't work with the markdown, so I'll try to describe it.

The essential idea is somewhat like a commutative diagram. Imagine a square, with the left top vertex A, right top vertex B, left bottom vertex A', and right bottom vertex B'. A is the original nonparametric regression problem (say, minimizing the MSE). B is the matrix version of the problem, which is essentially the same. B' is the robust matrix version of this problem proposed in this paper, and it is not directly related to the nonparametric regression problem A. A' is (if it exists) some robust type of nonparametric regression problem. It is related to A because it is the robust version, and it is related to B' because B' is the corresponding problem in the matrix space. Does A' exist? If so, what is it? The proposed solutions to B' should solve the (statistical) problem A'.

I think we should be open to accepting this paper because it may lead to interesting connections (A, A') and (A', B'), and there's room to explore here. Most borderline papers do not present this upside. Sure it may be forgotten as a dead end in a year or two, but that's most NeurIPS papers anyways.


**Time Spent Reviewing:**

3

---

> ### Author Response · Authors · 2021-08-10
> **Response to Reviewer cyV5**
>
> We thank the reviewer for your feedback.
>
> > I feel like there are some big unanswered questions. Why should we care about this particular formulation. Does this correspond to anything meaningful for a nonparametric regression problem? I know there are some short discussions on the particular divergences as related to normal random variables, but I don’t really understand what this is doing for my regression problem and why I should use it.
>
> Please see line 56-64, we discuss the motivation to use the reweighting approach for the weighted regression problem.
>
> - The non-parametric statistical estimator obtained by solving Problem (1) is sensitive to the corruptions of the training data. Similar phenomenon is also observed in machine learning where the solution of the risk minimization problem (1) is not guaranteed to be robust or generalizable.
>
> - The quality of the solution to Problem (1) also deteriorates if the training sample size $N$ is small.
>
> - Reweighting, obtained by modifying $\omega(z_i)$, is an attractive resolution to improve robustness and enhance the out-of-sample performance in the test data. At the same time, reweighting schemes have shown to produce many favorable effects: reweighting can increase fairness, and can also effectively handle covariate shift.
>
> However, it becomes non-trivial for the reweighting approach when the weight function is tied to a kernel (see line 65-74).
>
> - The kernel captures inherently the relative positions of the relevant covariates $z$, and any reweighting scheme should also reflect these relationships in a global viewpoint.
>
> - Another difficulty arises due to the lack of convexity or concavity, which prohibits the modification of the kernel parameters.
>
> - Modifying the covariate will also result in reweighting effects. Nevertheless, optimizing over the covariates is intractable for sophisticated kernels such as the Matern kernel.
>
> Our approach based on the adversarial reweighting schemes on doubly non-negative matrices can address this problem. At the same time, it also allows us to derive efficient algorithmic approaches based on first-order methods for the proposed framework.
>
> > Other issues that are more technical in nature get magnified by the fact that I don’t know what this robustification means for my regression problem. In Section 2, line 105, the discussion on the values of Ω^ is somewhat limited. On a first read of the paper, I thought the unspecified entries were 0, which would make the matrix positive semi-definite and therefore incompatible with the log-determinant divergence robustness. On taking another look, it became clear that one can pick the other entries of the matrix so as to make it positive semi-definite. But, almost certainly this choice is not unique. So, again, what does it do to my nonparametric regression problem to choose different matrices Ω^?
>
> We agree with the reviewer that the choice of $\hat \Omega$ is not unique. We, unfortunately, do not have any further insights on how the choice of $\hat \Omega$ may affect the estimator.
>
> As in the Problem (3), and the description for the nominal matrix of weights $\hat \Omega$ in line 103-106, the values of $\hat \Omega_{i, j}$ when $i \neq 0$ and $j \neq 0$ do not affect the objective and can be arbitrary. Following line 75-77 and line 121-124, we set the $\hat \Omega$ as the Gram matrix (See the function eval_Omega2 in our Matlab code, and the function nesterov_agd2 line 19 for more detail). Or, the $\hat \Omega_{i, j}$ when $i \neq 0$ and $j \neq 0$ is simply set to the kernel value between the $i^{th}$ and the $j^{th}$ data points for consistency.
>
> Thus, we would recommend to fill the matrix $\hat \Omega$ with the Gram matrix, which would be an intuitive and reasonable choice.

---

> > ### Comment · Area_Chair_cKbd · 2021-08-24
> > **Please respond to reviewer's post-rebuttal note**
> >
> > Dear Authors,
> >
> > Reviewer cyV5 had left a post-rebuttal note for you in their review, which I've copied below:
> >
> > "The essential idea is somewhat like a commutative diagram. Imagine a square, with the left top vertex A, right top vertex B, left bottom vertex A', and right bottom vertex B'. A is the original nonparametric regression problem (say, minimizing the MSE). B is the matrix version of the problem, which is essentially the same. B' is the robust matrix version of this problem proposed in this paper, and it is not directly related to the nonparametric regression problem A. A' is (if it exists) some robust type of nonparametric regression problem. It is related to A because it is the robust version, and it is related to B' because B' is the corresponding problem in the matrix space. Does A' exist? If so, what is it? The proposed solutions to B' should solve the (statistical) problem A'."
> >
> > It would be helpful  if you could send in a response to their question.
> >
> > Thanks,
> >
> > AC

---

> > > ### Author Response · Authors · 2021-08-24
> > > **Connection with robust estimators**
> > >
> > > We thank the reviewer for your consideration and for your kind appreciation of our paper. The reviewer edited the original comment, and we did not receive any email notification; we thank the Area chair for informing us about this edit.
> > >
> > > We first provide the correspondence between the reviewer’s vertices and the optimization problem in our paper:
> > >
> > > Vertex A ~ Problem (1)
> > >
> > > Vertex B ~ Problem (3)
> > >
> > > Vertex B’ ~ Problem (4)
> > >
> > > Problem A’ is obtained by projecting the set  $\mathcal U_{\varphi, \rho}(\hat{\Sigma})$ to the space of the weights, and then formulate a min-max optimization problem. In case the reviewer is interested in a precise formulation, problem A’ is
> > > $$
> > > 	\min\limits_{\beta}~\max\limits_{[\omega(\hat z_1),\ldots,\omega(\hat z_N)] \in \mathcal W} \sum_{i=1}^N \omega(\hat z_i) \ell (\beta, \hat x_i, \hat y_i),
> > > $$
> > > where the set $\mathcal W$ is defined as
> > > $$
> > > 	\mathcal W = \\{ [\omega(\hat z_1),\ldots,\omega(\hat z_N) ] \in \mathbb R_{+}^{N}: \exists \Omega \in \mathcal U_{\varphi, \rho}(\hat{\Sigma})  \text{ so that } \Omega_{0i} = \Omega_{i0} = \omega(\hat z_i) \forall i = 1, \ldots, N \\}.
> > > $$
> > >
> > > Thus, problem A’ is a min-max problem, which is also understood as a “robust estimation” problem.
> > >
> > > We understand that the reviewer may have a different viewpoint on what constitutes a “robust estimator”, and indeed, the field of robust statistics is also a vast area. To avoid any misunderstanding, let us define (in the context of this paper) that a robust estimator is an estimator obtained by solving a min-max optimization problem. This definition of robust estimator has been arising in the past few years, and has led to several insightful connections. For example, a min-max estimator is related to the variance-regularized estimator (arXiv:1610.02581), to the square-root LASSO estimator (arXiv:1610.05627), to the shrinkage estimator (arXiv:1805.07194), and so on. We understand that the reviewer is asking for this deep connection between problem A’ and some well-known statistical estimator or regularization schemes, unfortunately, we do not have a concrete answer to this open problem.
> > >
> > > In case the reviewer has any further questions, please add a comment so that we can answer in a timely manner. Thank you very much!

---

> > > > ### Comment · Reviewer_cyV5 · 2021-08-24
> > > > **not exactly what I meant**
> > > >
> > > > Apologies in advance for the terse response—I’m between meetings at the moment.
> > > >
> > > > To clarify, I don’t view Problem (1) as Vertex A. I view a standard non-parametric regression problem, e.g., minimize $\mathbb{E}[(f(X) - Y)^{2}]$ as the problem. I don’t really care about (1) if it doesn’t solve this or another problem of interest, but kernel estimators are well-established for solving this type of problem (see, e.g., Chapter 5 of A Distribution-Free Theory of Nonparametric Regression by Gyorfi et al.).
> > > >
> > > > I’m fine with a robust problem being a min-max, but it’s not obvious to me what the correct robust version of the statistical problem $\mathbb{E}[(f(X) - Y)^{2}]$ is in this setting. I think it’s fine for the matrix versions of the problem to be just optimization problems over data since we don’t care about the robust problem in matrix space anyways.

---

> > > > > ### Author Response · Authors · 2021-08-25
> > > > > **further clarifications**
> > > > >
> > > > > Thank you very much for your comment.
> > > > >
> > > > > To simplify the discussion, let us consider the case of the Nadaraya-Watson estimator in Example 1.1. In this case, the estimation problem is
> > > > >
> > > > > $$
> > > > > 	\min\limits_{\beta} \mathbb{E}_{\mathbb{P}} [ (\beta - Y)^{2} | Z = z_0] \quad \quad \quad (A).
> > > > > $$
> > > > > The above expectation requires the true data-generating distribution $\mathbb{P}$, which is unknown.
> > > > >
> > > > > The robustification used in our paper is obtained using the following two steps:
> > > > >
> > > > > *Step 1.* Use a kernel density estimator to form a nominal *conditional* probability measure of $Y | Z = z_{0}$. More precisely, this nominal distribution is a weighted sum of Dirac measures:
> > > > >
> > > > > $$
> > > > > 	 \hat{\mathbb{P}}\_{Y | Z = z\_0} (\mathrm{d} y)  \propto  \sum\_{i=1}^N K(z\_0, \hat z\_i)  \delta\_{\hat y\_i}(\mathrm{d} y),
> > > > > $$
> > > > >
> > > > > where $K$ is the kernel.
> > > > >
> > > > > *Step 2.* Solve the min-max problem
> > > > >
> > > > > $$
> > > > > \min\limits_{\beta} \max\_{\mathbb{Q}\_{Y | Z = z\_0} \in \mathcal{B}(\hat{\mathbb{P}}\_{Y | Z = z\_0} )} \mathbb{E}\_{\mathbb{Q}\_{Y | Z = z\_0}} [ (\beta - Y)^{2}] \quad \quad \quad (A’).
> > > > > $$
> > > > >
> > > > > The ambiguity set $\mathcal{B}(\hat{\mathbb{P}}\_{Y | Z = z\_0} )$ here is a set of *conditional* probability measures of $Y | Z = z\_0$, which is obtained by perturbing the weights of the Dirac measures in a specific way (projection of the matrix as we described previously).
> > > > >
> > > > > We hope that this could answer your question.

---

> > > > > > ### Comment · Reviewer_cyV5 · 2021-08-25
> > > > > > **further questions**
> > > > > >
> > > > > > Okay, so we’re getting closer. If you want to do consider the conditional distribution for (A) that’s fine—I think it makes more sense in this setting. Now, the (A’) you’ve given is still on the empirical distribution because of $\hat{\mathbb{P}}$, and for (A’) to be a robustification of (A), it should also be on the population conditional distribution.
> > > > > >
> > > > > > So, it really looks to me like you’re trying to solve a conditional (on z) distributionally robust optimization problem. I don’t know if that’s been studied yet (I imagine there must be a paper about it somewhere), but (1) you should try to find out and (2) you should try to characterize the nature of the robustness for the right (A’). Point (2) requires a better understanding of the projection of the uncertainty set, because that’s what we care about, not the uncertainty set itself. Does this projection correspond to a version of DRO that’s already been studied? Is it new?

---

> > > > > > > ### Comment · Area_Chair_cKbd · 2021-08-27
> > > > > > > **Re: further questions**
> > > > > > >
> > > > > > > Dear Authors,
> > > > > > >
> > > > > > > Thanks for the prompt and elaborate replies. The discussion phase would come to an end on September 2nd, and so, if you would like to send in a follow-up response to the reviewer, please do so in the next few days.
> > > > > > >
> > > > > > > If I understand correctly, the choice of the uncertainty set $\mathcal{U}$ results in a particular set of projected weights $\mathcal{W}$, based on which A' is defined. In your paper, you describe different ways to construct $\mathcal{U}$ (using two different divergence measures). I think the reviewer wants to know what these choice mean for the set $\mathcal{W}$, and for the original estimation problem of interest? Wouldn't an understanding of this connection be important to demonstrate the utility of your results to practical applications?
> > > > > > >
> > > > > > > Regards,
> > > > > > >
> > > > > > > AC

---

> > > > > > > > ### Author Response · Authors · 2021-08-30
> > > > > > > > **Further response to Reviewer cyV5 and Area Chair**
> > > > > > > >
> > > > > > > > We thank the Reviewer for the thoughtful comments, and the Area Chair for the elaboration.
> > > > > > > >
> > > > > > > > We followed the Reviewer’s guidance and below are our reply:
> > > > > > > >
> > > > > > > > **Literature**: we have checked more carefully the literature, and there are several recent papers on DRO for contextual decision making using various techniques such as residual-based (arXiv:2101.03139), trimmings of probability measures (arXiv:2009.10592), Wasserstein set (arXiv:2103.16451)... However, as far as we are aware of at this very moment, there is no paper that
> > > > > > > >
> > > > > > > > (i) builds the reference conditional measure $\hat{\mathbb{P}}\_{Y | Z = z\_0}$ using the kernel density,
> > > > > > > >
> > > > > > > > (ii) robustifies using a reweighting scheme,
> > > > > > > >
> > > > > > > > as we did in our paper.
> > > > > > > >
> > > > > > > > Following up, we believe that the projections we have (Bures-Wasserstein and Log-Determinant divergence) do **not** correspond to any existing DRO scheme that we know.
> > > > > > > >
> > > > > > > > We can visualize the projection set well for both the Bures-Wasserstein and the Log-Determinant divergence set if $N = 2$, unfortunately, we can’t include the plot in the response. Visualizing and grasping insights for the case when $N \ge 3$ is non-trivial.
> > > > > > > >
> > > > > > > > **Regarding your point (1) and (2)**: there are many statistical criteria for designing a robustification of (A’). Some of these include: (i) consistency, (ii) asymptotic optimality, (iii) convergence rate, (iv) finite sample guarantee, etc. Some of these properties may be trivially satisfied by our estimator. For example, the consistency of our estimator comes for free as $N \to \infty$ and $\rho \to 0$. Nevertheless, the big question is whether our estimator can have a better convergence rate, etc. Admittedly, these questions are not studied in this paper. So far, we have focused only on the design and computation of the new robustification schemes.
> > > > > > > >
> > > > > > > > **To answer the Area Chair’s comments**: We see our two schemes as **complementary** to one another. We do not see a rigorous way to justify theoretically that one approach is better than the other. A similar analogy is that there are many approaches to reduce overfitting (implicit/explicit regularization, robust optimization, adversarial training,...) and it is hard to clearly say ex-ante when one method is better than the others.
> > > > > > > >
> > > > > > > > A perfect use case for our methods, as elaborated by the Reviewer, is when we want to solve a conditional estimation problem, and we want to combine robustness and kernel. Other use cases are also mentioned briefly in Line 35--37, which include robustification for local learning methods, ERM with covariate shift, etc.
> > > > > > > >
> > > > > > > > Thank you very much!

---

### Official Review · Reviewer_DQfm · 2021-07-24

**Rating:** 7
**Confidence:** 2

**Summary:**

This paper focuses on reweighting the training samples in training a weighted regression model. To ameliorate the pessimism on the model's predictive power from low sample sizes or covariate perturbations, this paper proposes a new scheme for kernel-reweighted regression, i.e., adversarial reweighting scheme, to reparameterizing sample weights via a doubly non-negative matrix. Both theoretical and empirical analysis and results are provided.

**Limitations And Societal Impact:**

There is no foreseeable negative societal impacts.

**Main Review:**

Pros:

1. This paper is well organized.
2. Clear definition of the problem and related definition is provided, which allows an easy understand.
3. The proposed adversarial reweighting framework is of significance for kernel-reweighted regression.
4. Extensive experimental results and analysis on multiple real-world datasets are provided, which confirm the effectiveness and provide useful information about the proposed method.

Cons:

1. It seems to be a little bit confused that "While lifting the problem to the matrix space is not necessarily the most efficient approach, it endows us with more flexibility to perturb the weights in a coherent manner." (line 112 - 113). Could the author give a more intuitive explanation about why it needs to perturb the weights and why it is more flexible?
2. It is a little bit unclear about why it needs to be formulated into a min-max optimization problem. Could the author give more discussion about the motivation in the earlier part of this paper?
3. Lack of some intuitive explanation about how to address the potential problem results from low sample sizes or covariate perturbations. It may be better to add some discussion with the related properties in Sections. 3 or 4.

Minor question:

1. How to get the value of the perturbation intensity $\kappa$?


------Post-Rebuttal------

Thanks the authors for the detailed explanations.  After reading the author's response and other reviewers' comments, I think most of my concerns are well addressed. I decide to raise my score from 6 to 7.

Best,

Reviewer DQfm

**Time Spent Reviewing:**

8

---

> ### Author Response · Authors · 2021-08-10
> **Response to Reviewer DQfm**
>
> We thank the reviewer for your positive feedback.
>
> > It seems to be a little bit confused that "While lifting the problem to the matrix space is not necessarily the most efficient approach, it endows us with more flexibility to perturb the weights in a coherent manner." (line 112 - 113). Could the author give a more intuitive explanation about why it needs to perturb the weights and why it is more flexible?
>
> Please see line 17-18, in Problem (1), the weight function $\omega$ indicates the contribution of the sample-specific loss to the objective. Moreover, as in line 56-64, we discuss the motivation to use the reweighting approach for the weighted regression problem. However, it becomes non-trivial for the reweighting approach when the weight function is tied to a kernel (see line 65-74).
>
> Lifting to a matrix space is more flexible because:
>
> - (i) we have some flexibility to choose $\hat{\Omega}$,
>
> - (ii) the geometry of the matrix space is also richer (with the log-determinant divergence or the Bures-Wasserstein distance).
>
> > It is a little bit unclear about why it needs to be formulated into a min-max optimization problem. Could the author give more discussion about the motivation in the earlier part of this paper?
>
> The min-max formulation in this paper is motivated by the “robustifying” mechanisms in adversarial training and robust optimization community (see arXiv:1706.06083, arXiv:1710.10571, arXiv:1908.08729, and also the seminal book on Robust Optimization by Ben-Tal, El Ghaoui and Nemirovski). Thus we follow the convention of these (sub)fields and we call an estimator to be ``robust’’ if it is obtained by solving the min-max formulation (4). We will elaborate more on the motivation of the min-max formulation.
>
> > Lack of some intuitive explanation about how to address the potential problem results from low sample sizes or covariate perturbations. It may be better to add some discussion with the related properties in Sections. 3 or 4.
>
> Please see line 56-64, the reweighting is an effective approach to improve robustness and enhance the out-of-sample performance in the test data (also see the references [23, 26, 32]).
>
> We would also want to emphasize that the robust approach (via the min-max approach) typically performs well for low sample size because it is also a method to reduce overfitting in machine learning, see the introduction in arXiv:1908.08729 for a discussion.
>
> > How to get the value of the perturbation intensity $\kappa$?
>
> Our proposed approach is agnostic of $\kappa$ (it does not require the value of $\kappa$ in order to calculate the estimate). The perturbation intensity $\kappa$ is used to create multiplicative perturbations for the response only in the numerical experiment.

---

### Decision · Program_Chairs · 2021-09-28

**Decision:**

Accept (Poster)

**Comment:**

The paper proposes ways to robustify estimates for kernel-weighted regression. Most of the reviewers agree that the paper is technically sound and that it has the potential for interesting follow-up work, but there were concerns raised about the clarity and motivation, and absence of concrete connections between the robust reformulated problem and the original estimation problem. Having gone through the paper, I concur with the reviewers on the exposition issues pointed out, but agree that the paper should be accepted.

We strongly urge the authors to work on improving the exposition of the paper in the final version, and in particular:
- improve the motivation and description of the problem in the introduction
- and more importantly, have a discussion on the connections between the matrix re-parametrized robust problem (B' in Reviewer cyV5's comments) and the original estimation problem (A and A').

I thank the authors for engaging in an elaborate discussion with Reviewer cyV5. My reading is that there are two aspects that are unclear: (i) the choice of the uncertainty set and what it means for the original problem A, and (ii) if the resulting robust problem A' has connections to existing robustification schemes in the literature. I think the answer to the latter is "no", and it would be good to explicitly state this in the paper. One way to bring out of these nuances is by working out an end-to-end example (e.g. using the Nadaraya-Watson estimator), similar to what was presented in the rebuttal, but I leave it to authors to decide the best way to address the issues raised.

It's after much deliberation that we've decided to accept the paper. We've decided to give the author's the benefit of the doubt, and trust that they would bring out the same clarity in the paper as they did in the author response.

**Consistency Experiment:**

NeurIPS has a long history of experimentation. In 2014, NeurIPS ran an experiment in which 10% of submissions were reviewed by two independent committees to quantify the randomness in the review process. This year, we repeated a variant of this experiment to see how the quality of the review process has changed over time.  This paper was part of the experiment and was therefore assigned to two committees (consisting of reviewers, an Area Chair, and a Senior Area Chair) that reached independent decisions.  If both committees made the same recommendation, this recommendation was followed. If a single committee recommended acceptance, the paper was accepted (with the exception of a few cases in which the other committee identified what we considered a fatal flaw, e.g., an error in a key result).

This copy’s committee reached the following decision: **Accept (Poster)**

The other committee assigned to the paper recommended **Reject**.  You can find the other set of reviews, along with any follow up discussion with the authors here:
https://openreview.net/forum?id=npvEdo4Ftb1